  SciPost Phys. Lect.Notes 31 (2021)

# Bogoliubov quasiparticles in superconducting qubits

**Leonid I. Glazman[1] and Gianluigi Catelani[2,3]⋆**

**1** Departments of Physics and Applied Physics, Yale University,
New Haven, Connecticut 06520, USA
**2** JARA Institute for Quantum Information (PGI-11),
Forschungszentrum Jülich, 52425 Jülich, Germany
**3** Yale Quantum Institute, Yale University, New Haven, Connecticut 06520, USA

⋆ g.catelani@fz-juelich.de

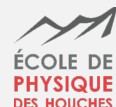

*Part of the Quantum Information Machines*
*Session 113 of the Les Houches School, July 2019*
*published in the Les Houches Lecture Notes Series*

## Abstract

Extending the qubit coherence times is a crucial task in building quantum information processing devices. In the three-dimensional cavity implementations of circuit QED, the coherence of superconducting qubits was improved dramatically due to cutting the losses associated with the photon emission. Next frontier in improving the coherence includes the mitigation of the adverse effects of superconducting quasiparticles. In these lectures, we review the basics of the quasiparticles dynamics, their interaction with the qubit degree of freedom, their contribution to the qubit relaxation rates, and approaches to control their effect.

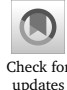

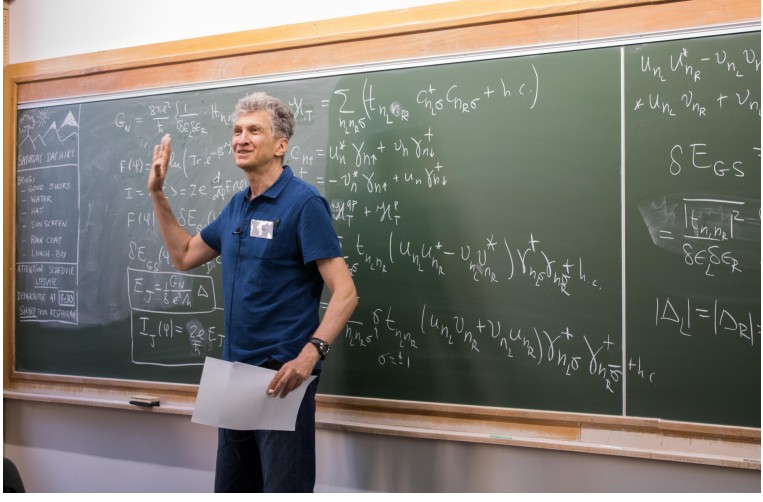

# 1 Superconductivity in an isolated metallic island

## 1.1 Electron pairing and condensate

Exposition of the theory of superconductivity conventionally starts with considering electrons freely propagating as plane waves in an ideal, translationally-invariant medium [1]. The electron energy spectrum is then continuous. The number of electron states per unit volume per unit energy, usually referred to as *the density of states*, is a function of energy, with some finite value $\nu_0$ at the Fermi level. To give a bit different perspective on the subject, let us consider, instead, a medium confined to some large (in units of Fermi wavelength) box and containing finite density of impurities which scatter electrons. Confinement to the box renders electron spectrum discrete, while scattering off randomly-positioned impurities would remove any accidental degeneracy of the levels. Under these conditions, the average *density of energy levels* $\xi_\alpha$ of one-electron states $\alpha$ in the vicinity of the Fermi energy is $\mathcal{V}\nu_0$, with $\mathcal{V}$ being the vol-

ume of the box. The typical spacing between the adjacent energy levels then is $\delta\epsilon = 1/\mathcal{V}\nu_0$. Taking for a crude estimate $\nu_0 = 1\,(\text{eV}\cdot\text{Å}^3)^{-1}$, we find for an island of volume $\mathcal{V} = 10^{-2}\,\mu\text{m}^3$ the average density of levels $10^{10}\,\text{eV}^{-1}$, yielding a tiny level spacing of $\delta\epsilon = 10^{-10}\,\text{eV} \approx 1\,\mu\text{K}$. Hereinafter the term "average" means average over an energy interval which includes many levels, but still very small compared to the Fermi energy $E_F$ (typically a few eV in a conventional metal). The Kramers theorem indicates that in the absence of magnetization each discrete electron level in a normal-metal island is doubly-degenerate, forming a Kramers pair. This statement is unaffected by the spin-orbit coupling, as it does not break the time-reversal symmetry. For simplicity, however, we will dispense with the spin-orbit coupling and associate the pair of states with the spin-up and spin-down electrons having the same orbital part of the wave function $\psi_n(\mathbf{r})$ (this is an excellent approximation for light elements, such as Al). In terms of these states, the second-quantized form of the Hamiltonian is (for brevity, we do not include the spin-triplet channel for the interaction which does not change the conclusions)

$$\mathcal{H} = \sum_{n,\sigma=\uparrow,\downarrow} \xi_n c_{n\sigma}^\dagger c_{n\sigma} + \sum_{klmn} \mathcal{H}_{klmn} c_{k\uparrow}^\dagger c_{l\downarrow}^\dagger c_{m\downarrow} c_{n\uparrow}. \tag{1}$$

Here operators $c_{n\sigma}^\dagger$ and $c_{n\sigma}$ create and annihilate electrons with energies $\xi_n$ (measured from the Fermi level), and

$$\mathcal{H}_{klmn} = \int d\mathbf{r}_1 d\mathbf{r}_2 V(\mathbf{r}_1 - \mathbf{r}_2)\psi_k^\star(\mathbf{r}_1)\psi_l^\star(\mathbf{r}_2)\psi_m(\mathbf{r}_2)\psi_n(\mathbf{r}_1) \tag{2}$$

are the matrix elements of interaction, written in terms of the single-particle eigenfunctions. These are strongly oscillating in space, and there is little correlation between the oscillations of the wavefunctions of different states. As a result, there is a strong hierarchy in the matrix elements $\mathcal{H}_{klmn}$: those with pairwise-equal indices are by far the largest ones. We will illustrate it using an example of a contact interaction, $V(\mathbf{r}) = (\lambda/\nu_0)\delta(\mathbf{r})$, characterised by dimensionless interaction constant $\lambda$. In this example, the double-integral in the right-hand side of Eq. (2) is reduced to an integral over a single variable $\mathbf{r}$ with the integrand $(\lambda/\nu_0)\psi_k^\star(\mathbf{r})\psi_l^\star(\mathbf{r})\psi_m(\mathbf{r})\psi_n(\mathbf{r})$. For generic $k,l,m,n$ the product of wave functions is rapidly oscillating as a function or $\mathbf{r}$ thus suppressing the value of the integral (the characteristic length scale for the oscillations is set by the Fermi wavelength) and making it zero on average. Having $k = n$, $l = m$ or $k = m$, $l = n$ reduces the product of wavefunctions to $|\psi_k(\mathbf{r})|^2|\psi_l(\mathbf{r})|^2$ which is non-negative, no matter if the one-particle wave functions real- or complex-valued. A non-negative integrand leads to matrix elements $\mathcal{H}_{kllk}$ and $\mathcal{H}_{klkl}$ which only weakly depend on $k$ and $l$, having non-zero average $\sim \lambda\delta\epsilon$. In the presence of a magnetic flux piercing the island wavefunctions the time-reversal symmetry is broken, and $\psi_k(\mathbf{r})$ are complex-valued. To the contrary, time-reversal symmetry allows one to choose real-valued eigenfunctions $\psi_n(\mathbf{r}) = \psi_n^\star(\mathbf{r})$. That brings yet another paring, $k = l$, $m = n$, yielding a non-negative product $[\psi_k(\mathbf{r})]^2[\psi_m(\mathbf{r})]^2$.

The said three types of parings correspond, respectively, to the Hartree term, Fock term, and the Bardeen-Cooper-Schrieffer (BCS) term. These three interaction types are the leading ones, regardless the details of $V(\mathbf{r})$, including its range and sign. Accounting for the Coulomb long-range component of $V(\mathbf{r})$ generates the charging energy out of the Hartree term, while the Fock term induces exchange interaction (which is safe to ignore in the case of a nonmagnetic material); these two interactions are insensitive to breaking the time-reversal symmetry. The BCS term is responsible for the formation of a superconducting state, once $V(\mathbf{r})$ contains a short-range attraction component. Therefore, neglecting the exchange interaction and level-to-level fluctuations, the interaction term in the island Hamiltonian Eq. (1) takes a universal form,

$$\mathcal{H}_{\text{int}} = E_C(\hat{N}^e - \mathcal{N}_g)^2 + (\lambda\delta\epsilon)\hat{O}^\dagger\hat{O}, \tag{3}$$

independent on the details of the electron wavefunctions in the island. Here

$$\hat{N}^e = \sum_{n,\sigma} c_{n\sigma}^\dagger c_{n\sigma} \tag{4}$$

is the operator of the number of electrons, and accounting for

$$\hat{O} = \sum_{|\xi_n| < \hbar\omega_D} c_{n\downarrow} c_{n\uparrow} \tag{5}$$

allows one to consider superconductivity in case of the attractive interaction between electrons with energies within some range $|\xi_n| < \hbar\omega_D$ (for the phonon mechanism of superconductivity, $\omega_D$ is of the order of phonon Debye frequency). The superconducting phase transition is associated with the appearance of a macroscopically-large value of $\langle \hat{O}^\dagger \hat{O} \rangle$ defeating the smallness of the factor $\lambda\delta\epsilon$ in Eq. (3).

The electron number $N^e$ is conserved in an isolated island, so the included in Eq. (3) polarization charge $\mathcal{N}_g$ for now reflects only the level from which all energies are measured. Now we consider fixed even $N^e$ and therefore fixed charging energy represented by the first term in Eq. (3), and concentrate on the ground state of an isolated superconducting island described by Hamiltonian

$$\mathcal{H}_{\text{sc}} = \sum_{n,\sigma} \xi_n c_{n\sigma}^\dagger c_{n\sigma} + (\lambda\delta\epsilon)\hat{O}^\dagger\hat{O}. \tag{6}$$

The term $\hat{O}^\dagger\hat{O}$ in Eqs. (3), (6) is the counterpart of the BCS interaction term conventionally written [1] in the basis of plane waves,

$$\hat{O}^\dagger\hat{O} = \sum_{|\xi_n| < \hbar\omega_D} c_{n\uparrow}^\dagger c_{n\downarrow}^\dagger \sum_{|\xi_m| < \hbar\omega_D} c_{m\downarrow} c_{m\uparrow} \Longleftrightarrow \left( \sum_{|\xi_\mathbf{k}| < \hbar\omega_D} c_{\mathbf{k}\uparrow}^\dagger c_{-\mathbf{k}\downarrow}^\dagger \right) \left( \sum_{|\xi_\mathbf{p}| < \hbar\omega_D} c_{-\mathbf{p}\downarrow} c_{\mathbf{p}\uparrow} \right). \tag{7}$$

Either side of Eq. (7) preserves the total electron number and describes coupling involving large number of singlet pairs: in the case of an island, a pair on a level $n$ is coupled with $\sim \hbar\omega_D/\delta\epsilon$ other pair states labelled by $m$. Such type of coupling provides a motivation for applying a mean-field treatment for determining the ground-state energy and thermodynamics of the system. In the mean-field approximation, one introduces the average,

$$\Delta = (\lambda\delta\epsilon)\langle\hat{O}\rangle = (\lambda\delta\epsilon) \sum_{|\xi_m| < \hbar\omega_D} \langle c_{m\downarrow} c_{m\uparrow} \rangle, \tag{8}$$

to replace the quartic term in Eq. (6) by a bilinear one. After the simplified Hamiltonian,

$$\mathcal{H}_{\text{BCS}} = \sum_{n,\sigma} \xi_n c_{n\sigma}^\dagger c_{n\sigma} + \Delta^* \sum_{|\xi_m| < \hbar\omega_D} c_{m\downarrow} c_{m\uparrow} + \Delta \sum_{|\xi_m| < \hbar\omega_D} c_{m\uparrow}^\dagger c_{m\downarrow}^\dagger - \frac{|\Delta|^2}{\lambda\delta\epsilon}, \tag{9}$$

is diagonalized, one evaluates the average $\langle\dots\rangle$ in the right-hand side of Eq. (8) in terms of $\Delta$, forming this way a self-consistency equation for $\Delta$. This routine for an island is essentially identical to the one for a bulk superconductor. The bilinear mean-field Hamiltonian is diagonalized by the Bogoliubov transformation,

$$c_{n\uparrow} = u_n^* \gamma_{n\uparrow} + v_n \gamma_{n\downarrow}^\dagger \tag{10}$$

$$c_{n\downarrow}^\dagger = -v_n^* \gamma_{n\uparrow} + u_n \gamma_{n\downarrow}^\dagger. \tag{11}$$

Here $\gamma_{n\sigma}^{\dagger}$, $\gamma_{n\sigma}$ are creation and annihilation operators for quasiparticle excitations with spin $\sigma = \uparrow, \downarrow$. The Bogoliubov amplitudes are complex numbers; for convenience, we may define gauge by taking $u_n = u_n^*$, $v_n = |v_n|e^{i\varphi}$ with $\varphi$ being the phase of the order parameter, $\Delta = |\Delta|e^{i\varphi}$. To preserve canonical commutation relations, their magnitudes satisfy the constraint

$$|v_n|^2 = 1 - |u_n|^2 = \frac{1}{2}\left(1 - \frac{\xi_n}{\epsilon_n}\right), \tag{12}$$

where $\epsilon_n = \sqrt{\xi_n^2 + |\Delta|^2}$ is the quasiparticle excitation energy. The ground state $|GS\rangle$ of the mean-field Hamiltonian is defined by the condition $\gamma_{n\sigma}|GS\rangle = 0$. The order parameter $\Delta$ is found self-consistently as

$$\Delta = (\lambda\delta\epsilon)\sum_n u_n^* v_n(1 - \langle\gamma_{n\uparrow}^{\dagger}\gamma_{n\uparrow}\rangle - \langle\gamma_{n\downarrow}^{\dagger}\gamma_{n\downarrow}\rangle). \tag{13}$$

Please note that the right-hand side here remains finite in the macroscopic limit $\delta\epsilon \to 0$. Finally, the Hamiltonian in Eq. (9) is transformed into the Hamiltonian for quasiparticle excitations:

$$\mathcal{H}_{\text{qp}} = \sum_{n,\sigma} \epsilon_n \gamma_{n\sigma}^{\dagger}\gamma_{n\sigma}. \tag{14}$$

Thermal averages present in Eq. (13) are evaluated over the Gibbs ensemble with the Hamiltonian Eq. (14). The self-consistency equation defines the absolute value of the order parameter $|\Delta(T)|$, leaving its phase $\varphi$ arbitrary; the ground-state energy, excitation spectrum, and thermodynamic potential of an island are independent of $\varphi$. The zero-temperature solution of Eq. (13) yields $|\Delta| \approx 2\hbar\omega_D \exp(-1/\lambda)$. It remains finite in the limit $\delta\epsilon \to 0$; for islands of a typical size, $\delta\epsilon \ll |\Delta| \ll \hbar\omega_D$. The ground-state wave function of the mean-field Hamiltonian (9) with a given $\Delta$ is

$$|\psi_\varphi\rangle = \prod_n \left(u_n + v_n c_{n\uparrow}^{\dagger}c_{n\downarrow}^{\dagger}\right)|0\rangle, \tag{15}$$

where $|0\rangle$ is the vacuum for electronic excitations, $c_{n\sigma}|0\rangle = 0$. The subscript $\varphi$ indicates that the phase of $\Delta$ enters this definition via the Bogoliubov amplitudes. One can verify that this expression satisfies the condition $\gamma_{n\sigma}|\psi_\varphi\rangle = 0$ defining the ground state for quasiparticle excitations (cf. Sec. 2.3). Clearly, the defined by Eq. (15) functions are $2\pi$-periodic: $|\psi_\varphi\rangle = |\psi_{\varphi+2\pi}\rangle$.

While being an eigenstate of the BCS Hamiltonian, $|\psi_\varphi\rangle$ is not an eigenfunction of the electron number. It is rather a coherent superposition of states with different numbers of electron pairs, so the number of pairs is not defined. The ground-state energy and the excitations spectrum are independent of $\varphi$, which provides a relief: out of $|\psi_\varphi\rangle$ functions, we may form a linear combination

$$|\psi_{N_P}\rangle = \int_0^{2\pi} \frac{d\varphi}{2\pi} e^{-iN_P\varphi}|\psi_\varphi\rangle \tag{16}$$

corresponding to a definite number of electron pairs $N_P$. The relation (16) is gauge-invariant (*i.e.*, invariant with respect to an arbitrary phase shift, $\varphi \to \varphi + \varphi_0$). The wave function (16) is an excellent approximation to the ground state of the Hamiltonian (6), which conserves the electron number. The associated with the finite level spacing $\delta\epsilon$ corrections to $\Delta$ of Eq. (13) and to the corresponding ground-state energy of Hamiltonian (6) scale to zero proportionally to $\delta\epsilon$ with the increase of island volume (see [2] for details and further references).

The condensate wavefunctions $|\psi_\varphi\rangle$ and $|\psi_{N_P}\rangle$ form two bases in the Hilbert space of many-body paired electron states. We may view the $N_P$ and $\varphi$ representations as dual ones, similar to $\hat{x}$ and $\hat{p}$ representations in the single-particle quantum mechanics. (The most important difference is that $\varphi$ varies between 0 and $2\pi$, making it a compact variable.) In the $N_P$

representation, the operator of number of electron pairs $\hat{N}$ (measured from some large integer corresponding to the filled Fermi sea in the island) acts as a multiplication operator, $\hat{N} = N\cdot$. Now we establish its form in the $\varphi$ representation:

$$N|\psi_N\rangle = N \int_0^{2\pi} d\varphi\, e^{-iN\varphi}|\psi_\varphi\rangle = \int_0^{2\pi} d\varphi \left( i\frac{d}{d\varphi} e^{-iN\varphi}\right)|\psi_\varphi\rangle \tag{17}$$
$$= \int_0^{2\pi} d\varphi\, e^{-iN\varphi}\left(-i\frac{d}{d\varphi}|\psi_\varphi\rangle\right).$$

That is, $\hat{N} = -id/d\varphi$. It is important to remember that the functions $|\psi_\varphi\rangle$ are $2\pi$-periodic, so the spectrum of $-id/d\varphi$ is the set of integers ($N = 0$ means no extra electron pairs on the island). Conversely, the operator

$$\hat{T} = \sum_N |N+1\rangle\langle N| \tag{18}$$

increasing the number of pairs by 1 is a multiplication operator in the $\varphi$-representation:

$$|\psi_{N+1}\rangle = \int_0^{2\pi} d\varphi\, e^{-i(N+1)\varphi}|\psi_\varphi\rangle = \int_0^{2\pi} d\varphi\, e^{-i\varphi}\left(e^{-iN\varphi}|\psi_\varphi\rangle\right) \tag{19}$$

so that $\hat{T} = e^{-i\varphi}$. Therefore in the space of states we considered here, variable $\hat{N}$ is a conjugate to the compact variable $\hat{\varphi}$. The two satisfy the appropriate canonical commutation relation

$$\left[\hat{N}, e^{-i\hat{\varphi}}\right] = e^{-i\hat{\varphi}}, \tag{20}$$

invariant with respect to the basis.

## 1.2  Thermodynamics of a superconducting island

The electron condensate in an isolated island accommodates an even number of particles. If the number of electrons on the island is even, they all reside in the condensate in a $T = 0$ equilibrium state. Under the same conditions, an odd electron in the island does not have a pair and occupies the lowest-energy quasiparticle state, thus raising the energy of the island by $|\Delta|$. At higher temperatures, ionization of the Cooper pairs results in a higher number of equilibrium quasiparticles, diminishing this even-odd effect. To see this we evaluate and compare the partition functions $Z_0$ and $Z_1$ for the even and odd numbers of electrons, respectively [3]. In the "even" case the states of the island are parametrized by the number $0, 2, 4, \ldots$ of quasiparticles and their quantum numbers, so we write:

$$Z_0 = 1 + \frac{1}{2!}\sum_{n_1,n_2}\exp\left(-\frac{\epsilon_{n_1}+\epsilon_{n_2}}{T}\right) + \frac{1}{4!}\sum_{n_1\ldots n_4}\exp\left(-\frac{\epsilon_{n_1}+\epsilon_{n_2}+\epsilon_{n_3}+\epsilon_{n_4}}{T}\right) + \ldots \tag{21}$$

(hereinafter we disregard the negligible probability of double-occupancy of any state). The series here is easy to sum up:

$$Z_0 = \cosh z(T, \delta\epsilon, \Delta), \tag{22}$$
$$z(T, \delta\epsilon, \Delta) = \sum_n \exp\left(-\frac{\epsilon_n}{T}\right) \simeq (\sqrt{2\pi T|\Delta|}/\delta\epsilon)e^{-|\Delta|/T}. \tag{23}$$

Similarly in the "odd" case the island contains $1, 3, 5, \ldots$ quasiparticles, and the partition function equals

$$Z_1 = \sum_n \exp\left(-\frac{\epsilon_n}{T}\right) + \frac{1}{3!}\sum_{n_1,n_2,n_3}\exp\left(-\frac{\epsilon_{n_1}+\epsilon_{n_2}+\epsilon_{n_3}}{T}\right) + \cdots = \sinh z(T, \delta\epsilon, \Delta). \tag{24}$$

One can easily recognize $n_{qp} = 2z(T, \delta\epsilon, \Delta)/\mathcal{V}$ as the quasiparticle density in the bulk at $T \ll \Delta$ (the factor of 2 accounting for spin). It is convenient to normalize $n_{qp}$ by the "density of Cooper pairs" $n_{CP}$,

$$n_{qp} = n_{CP}x_{qp}, \quad n_{CP} = 2\nu_0\Delta. \tag{25}$$

In equilibrium, $x_{qp} = \sqrt{2\pi T/|\Delta|}\exp(-|\Delta|/T)$.

The difference between the thermodynamic potentials of the even and odd states, namely $T\ln(Z_0/Z_1)$, becomes substantial (order-of-$\Delta$) once on average there is less than one thermally excited quasiparticle on an island, i.e., $z \lesssim 1$. This happens at $T$ below the scale set by the $\mathcal{V}$-dependent Cooper pair ionization temperature

$$T^\star = |\Delta|/\ln(n_{CP}\mathcal{V}) \approx |\Delta|/\ln(|\Delta|/\delta\epsilon). \tag{26}$$

The logarithm in the denominator here is pretty big, it is about 14 for an Al island of a typical volume $\mathcal{V} = 10^{-2}\mu m^3$. As a result, one expects - on the grounds of thermodynamics – no broken Cooper pairs in such island at $T \lesssim T^\star = 0.14$K. In our example, we find $x_{qp} \approx 1.9 \cdot 10^{-7}$ at $T = T^\star$ and may expect a minuscule $x_{qp} \approx 2.1 \cdot 10^{-23}$ at a typical for qubit experiments temperature $T = 40$mK. However, numerous measurements find $x_{qp} = 10^{-7} - 10^{-5}$ at these temperatures. The origin of the excess quasiparticles is not known and remains under scrutiny. Meanwhile, it is worth assessing how harmful they are for the qubits operation and look for ways to mitigate their unwanted effects.

## 2 Linking the islands

### 2.1 Josephson junctions phenomenology and a model of a single-junction qubit

Upon linking the islands, electrons may flow from one island to another. However, at energies low compared to the gap $|\Delta|$ in the excitations spectrum, only Cooper pairs facilitate the electron transfer. The corresponding Hamiltonian $\mathcal{H}_J$ of a link between two islands (L and R) therefore is a function of the products $\hat{T}_R^\dagger\hat{T}_L$ and $\hat{T}_L^\dagger\hat{T}_R$ [cf. Eq. (18)],

$$\mathcal{H}_J = \sum_{n=1}^{\infty}\left(C_n\hat{T}_R^{\dagger n}\hat{T}_L^n + C_n^*\hat{T}_L^{\dagger n}\hat{T}_R^n\right) + \text{const}. \tag{27}$$

This Hamiltonian captures the coherent, non-dissipative tunneling of pairs of electrons; each term of the sum corresponds to transfer of $n$ Cooper pairs in a single tunneling event. For a conventional tunnel junction, electron transmission coefficient is small, so one may safely keep only the lowest-order term ($n = 1$) in the sum. Furthermore, time-reversal symmetry for tunneling through a non-magnetic insulator dictates $C_1^* = C_1$. Using the phase representation $(\varphi_L, \varphi_R)$ for the operators $T_L$ and $T_R$ and omitting the phase-independent const term, we obtain

$$\mathcal{H}_J = -E_J\cos\varphi, \quad \varphi = \varphi_R - \varphi_L. \tag{28}$$

The connected islands at $\varphi = 0$ constrain the motion of a Cooper pair less than each island separately, so the ground-state energy of the entire system is reduced by the link; it means that in Eq. (27) the only remaining coefficient $C_1 < 0$, and therefore $E_J > 0$ in Eq. (28). We note in passing that the Josephson energy $E_J$ and the normal-state conductance of the junction $G$ do not have to be small compared, respectively, to $\Delta$ and $e^2/h$, as the smallness of transmission coefficient may be compensated by a large area of the junction. Later on, we will evaluate $E_J$ microscopically and relate it to $G$.

One more remark is due here: we tacitly assumed that the ground state of the system is non-degenerate. Tunneling via a quantum dot carrying an uncompensated electron spin provides

a counter-example, as the Kramers degeneracy is preserved at sufficiently weak tunneling [4]. The presence of the localized spin results in $E_J < 0$, so that the lowest energy of the junction is reached [4] at $\varphi = \pi$. Formation of a $\pi$-junction in tunneling through a quantum dot was demonstrated, *e.g.*, in Ref. [5].

Transfer of $N$ Cooper pairs across the junction creates a charge dipole between the islands. The corresponding electrostatic energy [cf. Eq. (3)] in terms of the operator $\hat{N} = (1/i)d/d\varphi$, reads

$$\mathcal{H}_C = 4E_C \left( \frac{1}{i} \frac{d}{d\varphi} - n_g \right)^2 . \tag{29}$$

Here charging energy $E_C = e^2/2C$ takes into account the junction capacitance as well as any capacitance shunting the junction; $n_g$ is the static charge (in units of $2e$) induced by a biasing gate, background charges, and unpaired electrons. Out of the three contributions only the first one is controllable; the two others fluctuate on some large time scale. The contribution stemming from unpaired electrons is discrete, and changes by $\pm 1/2$ upon a quasiparticle tunneling across the junction; the background charge may vary continuously.

A single-junction qubit is described by the Hamiltonian $\mathcal{H}_J + \mathcal{H}_C$ acting in the space of periodic functions, $\psi(\varphi) = \psi(\varphi + 2\pi)$. Clearly, its spectrum is discrete, non-equidistant, and depends on $n_g$ periodically with period 1. The $n_g$-dependence is detrimental for the qubit coherence, due to the uncontrolled variations of $n_g$. In transmons [6], the unwanted sensitivity to $n_g$ is countered by increasing the ratio $E_J/E_C$. At $E_J/E_C \gg 1$, one may separate the quantum dynamics of the phase difference $\varphi$ into small fluctuations around the minima ($\varphi = 2\pi n$ with integer $n$) and discrete phase slips by $\pm 2\pi$. The former correspond to the dynamics of an anharmonic oscillator having non-equidistant levels needed for a qubit operation. The latter brings the unwanted sensitivity of the levels ($\propto \delta\varepsilon_n \cos 2\pi n_g$) to the uncontrolled variations of $n_g$. The probability amplitude of a phase slip is exponentially small at $E_J/E_C \gg 1$, $\delta\varepsilon_n \propto \exp(-\sqrt{8E_J/E_C})$. This allows one to effectively suppress the influence of $n_g$ without affecting the qubit energy levels (the relative anharmonicity $\alpha_r$ scales as a power law of $E_C/E_J$, $\alpha_r \simeq \sqrt{E_C/8E_J}$, see Ref. [6]).

A wide variety of experiments demonstrated the prominence of discrete $\pm 1/2$ jumps (commonly referred to as $e$-jumps) in the spectrum of fluctuations of $n_g$. Its average value, $\overline{n}_g$, can be controlled by a gate electrode in a properly-designed transmon. There are two special values of the gate voltage for which $\overline{n}_g = 1/4$ or $3/4$, making the transmon energy levels insensitive to the $\pm 1/2$ jumps. To see this, let us consider, *e.g.*, $\overline{n}_g = 1/4$. An $e$-jump changes this initial value of $\overline{n}_g$ to $\overline{n}_g = 3/4$. Due to the periodicity of the spectrum with $\overline{n}_g$, the energy levels of the qubit Hamiltonians with $\overline{n}_g = 3/4$ and $\overline{n}_g = -1/4$ in Eq. (29) are identical to each other. Lastly, we may change $\varphi \to -\varphi$, as the Josephson energy Eq. (28) is even in $\varphi$; this change would return the charging energy Hamiltonian after an $e$-jump to its initial form prior to the jump ($\overline{n}_g = 1/4$). The data for the qubit transition frequency, accumulated over a large series of sequential measurements, clearly shows the reduced sensitivity to the $e$-jumps at the said special values of $\overline{n}_g$, see Fig. 1(a) and Fig. 1(b).

In a fluxonium qubit [7], the protection is achieved by shunting the junction with a high-inductance loop. The loop – superinductor – is actually a chain of $n_s \sim 100$ Josephson junctions with sufficiently large $E_J^s/E_C^s$ so that the phase slips probability amplitude in the superinductor is negligible. This allows one to dispense with the periodicity of $\psi(\varphi)$ function and approximate the Hamiltonian of the superinductor by

$$\mathcal{H}_I = \frac{1}{2} E_L (\hat{\varphi} - 2\pi \Phi_e/\Phi_0)^2 . \tag{30}$$

The inductive energy $E_L = E_J^s/n_s = (\Phi_0/2\pi)^2/L$, with $\Phi_0 = \pi\hbar c/e$ being the quantum of flux, accounts for the loop inductance $L$; we also allowed for an external flux $\Phi_e$ threading the loop

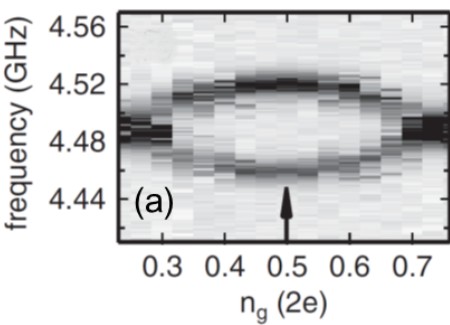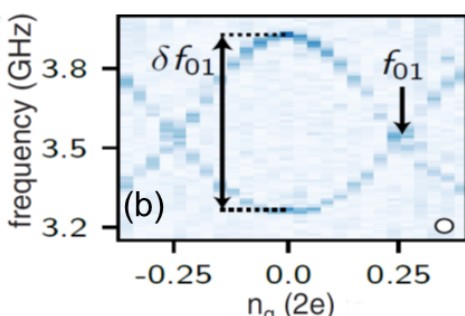

Figure 1: Panel (a) copied from Ref. [8]. Spectroscopy of a qubit as a function of gate-induced charge $n_g$. For each pixel, a Gaussian pulse ($\sigma = 20$ ns, corresponding to a pulse on resonance) is applied at the indicated frequency and the qubit is immediately measured. Each pixel is an average of 5000 repetitions (50 ms). Darker pixels correspond to higher homodyne readout voltages that are proportional to the probability of the qubit in the excited state. An "eye"-shaped pattern indicates charge-e jumps associated with the tunneling of nonequilibrium quasiparticles. Panel (b) copied from Ref. [9]. Normalized two-tone spectroscopy measurements of the $0 \to 1$ transition versus the offset charge.

formed by the superinductor and the qubit junction. The form of the Hamiltonian (30) dictates the boundary conditions for the wave function $\psi(\varphi \to \pm\infty) = 0$. This, in turn, allows one to eliminate $n_g$ from the Hamiltonian $\mathcal{H}_J + \mathcal{H}_I + \mathcal{H}_C$, by a simple gauge transformation. The inclusion of the shunt makes the energy levels of this Hamiltonian independent of $n_g$, while allowing for their control by $\Phi_e$.

To summarize, Hamiltonian

$$\mathcal{H}_\varphi = 4E_C \left( \frac{1}{i} \frac{d}{d\varphi} - n_g \right)^2 - E_J \cos\hat{\varphi} + \frac{1}{2} E_L (\hat{\varphi} - 2\pi\Phi_e/\Phi_0)^2, \tag{31}$$

describes a wide variety of superconducting qubits (see Fig. 2). In the absence of the superinductor ($E_L = 0$) Hamiltonian (31) acts in the space of periodic functions; if $E_L \neq 0$ then the proper boudary condtion is $\psi(\pm\infty) = 0$, which allows one to gauge out the $n_g$-dependence. Hamiltonian (31) acts in the low-energy subspace, meaning that it is good for describing energy levels well below the quasiparticle continuum; that, in turn, sets the requirement $E_J, E_C, E_L \ll \Delta$.

In metallic islands, screening length is very short (it is about the interatomic distance) and the energy of plasmon is extremely high (typically of the order of Fermi energy). As a result, quantum fluctuations of charge $\hat{N}$ governed by Eq. (31) lead merely to the fluctuations of the potential of an entire island; the associated with the fluctuations electric fields do not penetrate the bulk of an island. Fluctuations of the potential are benign for the gauge-invariant Hamiltonian (6) and do not affect its excited states and their occupations. Therefore, as long as quasiparticles do not tunnel and thus are not exposed to the *potential difference* between the islands, their presence is inconsequential to the dynamics of the qubit degree of freedom $\hat{\varphi}$. To elucidate the interaction of quasiparticles with $\hat{\varphi}$ we need to go beyond the phenomenology of Eqs. (27) and (28).

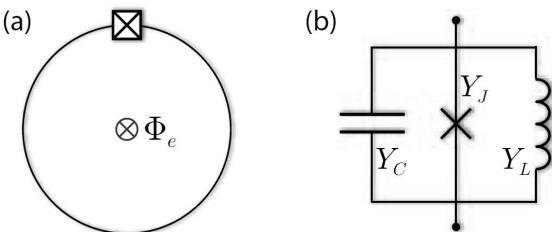

Figure 2: (from Ref. [10]) (a) Schematic representation of a qubit controlled by a magnetic flux, see Eq. (31). (b) Effective circuit diagram with three parallel elements – capacitor, Josephson junction, and inductor – characterized by their respective admittances.

## 2.2 Tunneling Hamiltonian and the normal-state conductance of a junction

Consider two normal-state leads separated by an insulating barrier. Electrons can tunnel through the barrier, and we model this system using the Hamiltonian:

$$\mathcal{H} = \mathcal{H}_L + \mathcal{H}_R + \mathcal{H}_T, \tag{32}$$

where $\mathcal{H}_\alpha$, $\alpha = L, R$ are the Hamiltonians of the left/right lead, and

$$\mathcal{H}_T = \sum_{n_L, n_R, \sigma} \left( t_{n_L n_R} c^\dagger_{n_L \sigma} c_{n_R \sigma} + \text{h.c.} \right) \tag{33}$$

is the tunneling Hamiltonian, describing transfer of an electron from a state $n_R$ in the right lead to a state $n_L$ in the left one, and a transfer in the opposite direction; $t_{n_L n_R}$ is the corresponding tunneling matrix element. If voltages $V_\alpha$ are applied to the leads, their Hamiltonians take the form

$$\mathcal{H}_\alpha(V_\alpha) = \mathcal{H}_\alpha - eV_\alpha N^e_\alpha, \tag{34}$$

where $\mathcal{H}_\alpha$ and the number operators $N^e_\alpha$ are given by the proper generalizations of Eqs. (1) and (4) to two leads. In the absence of tunneling, the particle number is conserved separately for the two leads, $[\mathcal{H}_\alpha, N^e_\alpha] = 0$. Tunneling allows the current to flow through a voltage-biased junction. This current is dissipative in the case of normal leads. There are many ways to evaluate it, below we present one of them.

It is convenient to change the representation, so that the excitation energies are measured from the respective electrochemical potential of each lead. For a generic, time-dependent unitary transformation $U(t)$, the transformed Hamiltonian $\tilde{\mathcal{H}}$ is given by

$$\tilde{\mathcal{H}} = U^\dagger \mathcal{H} U - i\hbar U^\dagger \frac{\partial U}{\partial t}. \tag{35}$$

Here we take $U$ in the form

$$U(t) = \exp\left[ i\phi_L(t) N^e_L + i\phi_R(t) N^e_R \right], \quad \phi_{L,R}(t) = \frac{e}{\hbar} \int^t V_{L,R}(t') dt' + \phi_{L,R} \tag{36}$$

(the specific value of constants $\phi_{L,R}$ here is inconsequential). Clearly, $U$ commutes with the lead Hamiltonians, $[\mathcal{H}_\alpha(V_\alpha), U] = 0$, while $[\mathcal{H}_T, U] \neq 0$. In the new representation, we have $\tilde{\mathcal{H}}_\alpha = \mathcal{H}_\alpha(V_\alpha = 0)$, $\tilde{N}^e_\alpha = N^e_\alpha$, and

$$\tilde{\mathcal{H}}_T = \sum_{n_L, n_R, \sigma} \left( t_{n_L n_R} e^{i[\phi_L(t) - \phi_R(t)]} c^\dagger_{n_L \sigma} c_{n_R \sigma} + t^*_{n_L n_R} e^{-i[\phi_L(t) - \phi_R(t)]} c^\dagger_{n_R \sigma} c_{n_L \sigma} \right). \tag{37}$$

A constant-in-time bias $V = V_L - V_R$ leads to $\phi_L(t) - \phi_R(t) = eVt/\hbar$, and the tunnel Hamiltonian takes form

$$\tilde{\mathcal{H}}_T = \sum_{n_L, n_R, \sigma} \left( t_{n_L n_R} e^{ieVt/\hbar} c^\dagger_{n_L \sigma} c_{n_R \sigma} + t^*_{n_L n_R} e^{-ieVt/\hbar} c^\dagger_{n_R \sigma} c_{n_L \sigma} \right). \tag{38}$$

In the used representation, this is a periodic-in-time perturbation of the Hamiltonian $\tilde{\mathcal{H}}_L + \tilde{\mathcal{H}}_R$, which results in the absorption of energy quanta $\hbar\Omega = eV$ by the system. To find the absorption power $P$ in the weak-tunneling limit, we may use the Born approximation for the tunneling amplitudes and then apply Fermi's Golden Rule to evaluate the transition rate; lastly, we multiply it by the energy quantum $eV$ to obtain:

$$P = \frac{4\pi}{\hbar} eV \sum_{n_L, n_R} |t_{n_L, n_R}|^2 \left[ f_F(\xi_{n_L}) - f_F(\xi_{n_R}) \right] \delta(\xi_{n_L} - \xi_{n_R} + eV). \tag{39}$$

We replaced the averages of $c^\dagger_n c_m$ over the Gibbs distributions with the Hamiltonians $\mathcal{H}_{L,R}$ by the respective Fermi functions $f_F(\xi_{n_L, n_R})$. This required neglecting the electron-electron interaction in the leads, which is fine for the conventional tunnel junctions between normal-state metallic electrodes; the extra factor of 2 in Eq. (40) accounts for the summation over the spin variable. One may worry that we applied a formalism developed for isolated islands to two leads attached to a voltage source. Indeed, the charge transfer in the DC regime between two otherwise isolated islands is not sustainable over an arbitrarily long time. On a technical level, the difficulty arises in the derivation of Fermi's Golden Rule, which we used in Eq. (39), for the transitions between the discrete spectrum levels, as one would end up with inter-level Rabi oscillations instead. A formal way around this difficulty is to consider a slightly broadened tone $\Omega$ which would compensate for the discreteness of the energies $\xi_{n_{L,R}}$ and allow one to use the standard derivation [11] of the Fermi's Golden Rule. (That derivation involves the consideration of the initial regime of linear growth of the perturbation [11]). After that, one takes the limit $\mathcal{V}_{L,R} \to \infty$ keeping the products $\mathcal{V}_L \mathcal{V}_R |t_{n_L, n_R}|^2$ finite, and then returns to a fixed $\hbar\Omega = eV$. To proceed with the derivation of DC conductance $G_N$, we use the relation $P = G_N V^2$ to find

$$G_N = \frac{4\pi e^2}{\hbar} \nu_L \nu_R \mathcal{V}_L \mathcal{V}_R \overline{|t_{n_L, n_R}|^2} = \frac{4\pi e^2}{\hbar} \frac{\overline{|t_{n_L, n_R}|^2}}{\delta\epsilon_L \delta\epsilon_R} \tag{40}$$

(we prefer the latter form of this relation for its compactness). In derivation of Eq. (40) we also assumed low temperature, $T \ll E_F$ and denoted an average over the states $n_L, n_R$ close to Fermi energy by $\overline{|t_{n_L, n_R}|^2}$. In an alternative standard derivation of Eq. (40), one uses momentum eigenstates for two clean infinite-size leads, see *e.g.* [12].

The microscopic properties of the junction are encoded in the matrix elements $t_{n_L, n_R}$ of the tunneling Hamiltonian and abbreviated to a single parameter, conductance, by Eq. (40). The same value of $G_N$ can be achieved by enlarging the cross-sectional area $\Sigma$ of the junction or by increasing the transmission coefficient $|t_B|^2$ of the tunnel barrier. The conductance of a large-area junction can be estimated as $G_N \sim (e^2/h)(\Sigma/\lambda_F^2)|t_B|^2$. The second factor here represents the cross-sectional area of the junction in units of the Fermi-wavelength-squared and can be viewed as the (large) number of independent electron modes fitting into the junction's area. Equation (40) will allow us to express various quantities of interest in the superconducting state in terms of the normal conductance $G_N$.

## 2.3 Josephson energy and current

We define current operator $\hat{I}$ as a time derivative of $eN_R^e$,

$$
\hat{I} = e\frac{dN_R^e}{dt} = e\frac{d\tilde{N}_R^e}{dt} = -\frac{ie}{\hbar}\left[\tilde{N}_R^e, \tilde{\mathcal{H}}_T\right] \tag{41}
$$
$$
= \frac{ie}{\hbar}\sum_{n_L,n_R,\sigma}\left(t_{n_L n_R}e^{i[\phi_L(t)-\phi_R(t)]}c_{n_L\sigma}^\dagger c_{n_R\sigma} - t_{n_L n_R}^* e^{-i[\phi_L(t)-\phi_R(t)]}c_{n_R\sigma}^\dagger c_{n_L\sigma}\right).
$$

At zero bias, the phase difference entering in Eq. (41) is independent of time, $\phi_L(t) - \phi_R(t) = \phi_L - \phi_R$, cf. Eq. (36). We may introduce the "superconducting" phase difference $\varphi = 2(\phi_L - \phi_R)$ and, by inspecting Eqs. (37) and (41), establish the relation $\hat{I} = (2e/\hbar)\partial\tilde{\mathcal{H}}_T/\partial\varphi$ between the current operator and tunneling Hamiltonian. Next we may account for $\tilde{\mathcal{H}}_L$ and $\tilde{\mathcal{H}}_R$ being independent of $\varphi$, in order to arrive at the exact relation, $\hat{I} = (2e/\hbar)\partial\tilde{\mathcal{H}}/\partial\varphi$, between the current operator and the full Hamiltonian (35). In equilibrium, i.e., with no bias applied, we may average the latter relation over the Gibbs distribution for the entire system and find for the current

$$
I \equiv \langle\hat{I}\rangle = \frac{2e}{\hbar}\frac{d}{d\varphi}F(\varphi), \quad F(\varphi) = -T\ln\mathrm{Tr}\,e^{-\mathcal{H}(\varphi)/T}. \tag{42}
$$

We removed the tilde sign in Eq. (42), as the trace there is gauge-invariant. A finite phase difference $\varphi$ for the electron states across the barrier may be introduced by including the junction in a conducting loop and threading a magnetic flux through it. This causes a current running in the loop in equilibrium (known as persistent current) even if the entire system is in the normal state. This mesoscopic effect, however, vanishes in the limit $\delta\epsilon \to 0$ and turns out quite hard to measure in normal-metal rings [13]. On the contrary, in the superconducting state, the Josephson current (42) remains finite at $\delta\epsilon \to 0$, i.e., in the limit of macroscopic leads.

We are interested in the Josephson current $I_J(\varphi)$ and energy at temperatures $T \ll \Delta$, so we may replace the free energy $F(\varphi)$ with the $\varphi$-dependent part $\delta E_G(\varphi)$ of the ground-state energy. We will evaluate $\delta E_G$ perturbatively to the lowest non-vanishing order $(t_{n_L,n_R}^2)$ in tunneling and within the BCS mean field approximation. In the "tilde" basis the phase dependence is delegated to $\tilde{\mathcal{H}}_T$, so the defined in Eq. (8) mean-field order parameters of the leads are purely real, $\Delta \to |\Delta_{L,R}|$. Alternatively, we may use the original basis, in which case the $\varphi_{L,R}$ dependencies are carried by $\Delta_{L,R}$. For definiteness, we use the latter gauge.

We now write the tunneling Hamiltonian $\mathcal{H}_T$, Eq. (33), in terms of the quasiparticle operators; using the Bogoliubov transformation, Eq. (10), we find

$$
\mathcal{H}_T = \mathcal{H}_T^{\mathrm{qp}} + \mathcal{H}_T^{\mathrm{p}} \tag{43}
$$
$$
\mathcal{H}_T^{\mathrm{qp}} = \sum_{n_L,n_R,\sigma} t_{n_L n_R}\left(u_{n_L}u_{n_R}^* - v_{n_L}v_{n_R}^*\right)\gamma_{n_L\sigma}^\dagger\gamma_{n_R\sigma} + \mathrm{h.c.} \tag{44}
$$
$$
\mathcal{H}_T^{\mathrm{p}} = \sum_{n_L,n_R,\sigma} \sigma t_{n_L n_R}\left(u_{n_L}v_{n_R} + v_{n_L}u_{n_R}\right)\gamma_{n_L\sigma}^\dagger\gamma_{n_R\bar{\sigma}}^\dagger + \mathrm{h.c.}, \tag{45}
$$

where for simplicity we assumed no bias ($V = 0$), and the presence of time-reversal symmetry resulting in real-valued tunneling amplitudes, $t_{n_L n_R} = t_{n_L n_R}^*$; we also adjusted the gauge, multiplying $u_n$ and $v_n$ by $e^{-i\varphi/2}$ compared to the definitions in Section 1.1. Here we have separated the terms accounting for single quasiparticle tunneling, $\mathcal{H}_T^{\mathrm{qp}}$, from that describing pair breaking and recombination, $\mathcal{H}_T^{\mathrm{p}}$. It is convenient to show explicitly the phase dependence of

the Bogoliubov amplitudes:

$$u_{n_L} u_{n_R}^* - v_{n_L} v_{n_R}^* = \left| u_{n_L} u_{n_R}^* \right| e^{i\varphi/2} - \left| v_{n_L} v_{n_R}^* \right| e^{-i\varphi/2}$$
$$u_{n_L} v_{n_R} + v_{n_L} u_{n_R} = \left| u_{n_L} v_{n_R} \right| e^{i\varphi/2} + \left| v_{n_L} u_{n_R} \right| e^{-i\varphi/2}, \tag{46}$$

where $\varphi = 2(\phi_L - \phi_R)$.

Obviously, the zeroth-order average $\langle \mathcal{H}_T \rangle_0 = 0$; in the next order one finds

$$\delta E_{GS} = -\sum_\lambda \frac{|\langle \lambda | \mathcal{H}_T | GS \rangle|^2}{E_\lambda}, \tag{47}$$

where the sum is over all possible excited states $|\lambda\rangle$ of energy $E_\lambda$ determined from the proper generalization of quasiparticle Hamiltonian Eq. (14) onto two leads. The only non-zero contribution comes from the first term in the RHS of Eq. (45). Then $|\lambda\rangle = |n_L n_R \sigma\rangle \equiv \gamma_{n_L \sigma}^\dagger \gamma_{n_R \bar{\sigma}}^\dagger |GS\rangle$ and $E_\lambda = \epsilon_{n_L} + \epsilon_{n_R}$. Evaluation of Eq. (47) yields:

$$\delta E_{GS} = -E_0(\Delta_L, \Delta_R) - E_J(\Delta_L, \Delta_R) \cos \varphi, \tag{48}$$
$$E_0 = \sum_{n_L, n_R} \frac{|t_{n_L n_R}|^2}{\epsilon_{n_L} + \epsilon_{n_R}} \left[ 1 - \frac{\xi_{n_L}}{\epsilon_{n_L}} \frac{\xi_{n_R}}{\epsilon_{n_R}} \right], \quad E_J = \sum_{n_L, n_R} \frac{|t_{n_L n_R}|^2}{\epsilon_{n_L} + \epsilon_{n_R}} \frac{|\Delta_L|}{\epsilon_{n_L}} \frac{|\Delta_R|}{\epsilon_{n_R}}.$$

Trading the summations here for the integration over the energies over the corresponding states and dispensing with the dependence of the tunneling matrix elements on $n_L, n_R$ we find

$$E_0(\Delta_L, \Delta_R) = 4 \frac{\overline{|t_{n_L n_R}|^2}}{\delta \epsilon_L \delta \epsilon_R} \int_{\Delta_L} \int_{\Delta_R} \frac{d\epsilon_L d\epsilon_R}{\epsilon_L + \epsilon_R} \frac{\epsilon_L \epsilon_R}{\sqrt{\epsilon_L^2 - |\Delta_L|^2} \sqrt{\epsilon_R^2 - |\Delta_R|^2}}, \tag{49}$$

$$E_J(\Delta_L, \Delta_R) = 4 \frac{\overline{|t_{n_L n_R}|^2}}{\delta \epsilon_L \delta \epsilon_R} \int_{\Delta_L} \int_{\Delta_R} \frac{d\epsilon_L d\epsilon_R}{\epsilon_L + \epsilon_R} \frac{|\Delta_L||\Delta_R|}{\sqrt{\epsilon_L^2 - |\Delta_L|^2} \sqrt{\epsilon_R^2 - |\Delta_R|^2}}. \tag{50}$$

The latter simplification made the integral in Eq. (49) ultraviolet-divergent; the dependence of $t$ on $n_L$ and $n_R$ provides one with a model-dependent cut-off at some energies of the order of $E_F$. The meaningful in the context of superconductivity part of $E_0(\Delta_L, \Delta_R)$, however, is model-independent. It can be evaluated with the help of the following regularization,

$$E_0(\Delta_L, \Delta_R) \to E_0(\Delta_L, \Delta_R) - E_0(0, 0),$$

which makes the integral convergent at $\epsilon_{L,R} \sim \Delta_{L,R}$ and provides a way to express const in Eq. (27) in terms of $\Delta_{L,R}$ and $G_N$. The integral in the Josephson energy (50) is convergent. Its evaluation at $|\Delta_L| = |\Delta_R| \equiv \Delta$ yields

$$\delta E_{GS} = -4 \frac{\overline{|t_{n_L n_R}|^2}}{\delta \epsilon_L \delta \epsilon_R} \Delta \cos \varphi \int_1 dx \int_1 dy \frac{1}{\sqrt{x^2 - 1}} \frac{1}{\sqrt{y^2 - 1}} \frac{1}{x + y}$$
$$= -\pi^2 \frac{\overline{|t_{n_L n_R}|^2}}{\delta \epsilon_L \delta \epsilon_R} \Delta \cos \varphi. \tag{51}$$

The applicability of the perturbation theory used in the derivation of Eq. (51) requires the smallness of the matrix elements $t_{n_L n_R}$, which in turn means that the transmission coefficient $|t_B|^2$ across the tunnel barrier must be small. As we discussed at the end of Section 2.2, the small factor $|t_B|^2 \ll 1$ in the conductance $G_N$ can be compensated by a large number of electron modes $\Sigma/\lambda_F^2$ fitting in the junction's cross-sectional area. A similar compensation

happens also for the Josephson energy Eq. (51). In fact, it can be expressed in terms of $G_N$ and $\Delta$. Comparing Eq. (51) with Eq. (40) and using Eq. (42) at $T \to 0$ we conclude:

$$\delta E_{GS}(\varphi) = -E_J \cos\varphi \,, \quad E_J = \frac{G_N}{G_Q}\frac{\Delta}{8} \,, \quad G_Q = \frac{e^2}{h} \,; \quad I_J(\varphi) = \frac{2e}{\hbar}\frac{d}{d\varphi}\delta E_{GS}(\varphi). \tag{52}$$

The expression of $E_J$ in terms of $\Delta$ and $G_N$ may be viewed as a version of the Ambegaokar-Baratoff formula. We may convert it to the familiar [1] form, $eI_c R_N = (\pi/2)\Delta$, by introducing the critical current of the junction $I_c = \max_\varphi\{I_J(\varphi)\}$ and its resistance $R_N = 1/G_N$ in the normal state.

## 2.4 Real part of the AC admittance of a junction

Josephson energy $E_J$ is one of the main parameters defining the energy spectrum of an ideal qubit, see Eq. (31). In the previous Section, we related $E_J$ to the normal-state conductance of the junction. Dissipation in the Josephson junction is one of the factors limiting the qubit coherence. To quantify the dissipation, this Section develops the theory of the AC admittance of a junction; we will focus on its real part.

Similar to our discussion of the DC conductance in Section 2.2, the shortest way to get the dissipative part of AC admittance, $\mathrm{Re}\, Y(\omega, \varphi)$, is to evaluate the absorption power $P$ of bias $V(t) = V\cos(\omega t)$ applied across the junction. In contrast to a fixed bias, the alternating one does not cause winding of the phase difference $\phi_L(t) - \phi_R(t)$ in Eq. (37). Instead, this difference exhibits small oscillations around a finite value, $\phi_L(t) - \phi_R(t) = \varphi/2 - eV\sin(\omega t)/\hbar\omega$. The smallness of $V$ allows us to expand $\tilde{\mathcal{H}}_T = \tilde{\mathcal{H}}_T(\varphi) - [2eV\sin(\omega t)/\hbar\omega]\partial_\varphi\tilde{\mathcal{H}}_T(\varphi)$ to the linear order in $V$. The oscillatory term in the expansion drives the transitions in which energy quanta $\hbar\omega$ are absorbed. For finding $P$ in the second order in $V$ and second order in $t_{n_L n_R}$ we may apply Fermi's Golden Rule to the perturbation $-[2eV\sin(\omega t)/\hbar\omega]\partial_\varphi\tilde{\mathcal{H}}_T(\varphi)$ and proceed similar to the derivation of Eq. (39). Next, we find the dissipative part of admittance by casting the result of calculation in the form

$$P = \frac{1}{2}\mathrm{Re}\left[Y(\omega; \varphi)\right]V^2. \tag{53}$$

This program works equally well for the tunnel junctions with normal or superconducting leads. In the former case, the result is independent of $\varphi$. We find $\mathrm{Re}\, Y(\omega) = G_N$ for a junction between two normal leads with energy-independent electron density of states, cf. Eq. (40). The limits $\omega \to 0$ and $V \to 0$ of the absorbed power do not commute in general; this is exemplified by the evaluation of the DC dissipative conductance of an SNS or Josephson junction [14]. In the following we concentrate on the case of a finite $\omega$.

Considering superconducting leads, we use Eqs. (43)-(45) in order to derive the appropriate form of the perturbation $-[2eV\sin(\omega t)/\hbar\omega]\partial_\varphi\tilde{\mathcal{H}}_T(\varphi)$ in terms of the quasiparticles creation and annihilation operators. This is achieved by replacing $u_{n_L} \to u_{n_L}\exp[i\varphi/2 + ie\int^t dt'V(t')/\hbar]$, $v_{n_L} \to v_{n_L}\exp[-i\varphi/2 - ie\int^t dt'V(t')/\hbar]$ in Eq. (44); the gauge invariance of the observables allows us to assign all the phase dependence to lead $L$. The operator structure of the $\mathcal{H}_T^p$ and $\mathcal{H}_T^{qp}$ parts of the perturbation is quite different: the former one describes creation or annihilation of pairs of quasiparticles, while the latter term corresponds to the quasiparticle tunneling. Therefore, absorption processes originating in $\mathcal{H}_T^p$ are effective only at frequencies exceeding the threshold, $\hbar\omega > 2\Delta$. Ultimately, we are interested in the interaction of quasiparticles with the qubit degrees of freedom evolving with

frequencies well below $\Delta/\hbar$. Therefore, we focus on the terms in $P$ stemming from $\mathcal{H}_T^{\mathrm{qp}}$,

$$\delta\mathcal{H}(t) = -[2eV\sin(\omega t)/\hbar\omega]\partial_\varphi\tilde{\mathcal{H}}_T^{\mathrm{qp}}(\varphi) = \mathcal{H}_{AC}\left(e^{i\omega t} - e^{-i\omega t}\right), \tag{54}$$

$$\mathcal{H}_{AC} = \frac{eV}{2\hbar\omega}\sum_{n_L,n_R,\sigma} t_{n_L n_R}\left(u_{n_L}u_{n_R}^* + v_{n_L}v_{n_R}^*\right)\gamma_{n_L\sigma}^\dagger\gamma_{n_R\sigma} - \mathrm{h.c.} \tag{55}$$

Power $P$ then is found by multiplying the energy quantum by the transition rate:

$$P = 2\pi\omega\sum_\lambda\langle\!\langle|\langle\lambda|\mathcal{H}_{AC}|\eta\rangle|^2\left[\delta(E_\lambda - E_\eta - \hbar\omega) - \delta(E_\lambda - E_\eta + \hbar\omega)\right]\rangle\!\rangle_{\mathrm{qp}}.$$

Double angular brackets denote averaging over the initial (not necessarily equilibrium) quasi-particles states $|\eta\rangle$ with energy $E_\eta$; sum is over possible final quasiparticle (qp) states $|\lambda\rangle$ with energy $E_\lambda$. The two terms in square brackets account for energy absorbed and emitted by quasiparticles, respectively. Performing the averaging and using Eqs. (40) and (53) we find for the dissipative admittance of the Josephson junction:

$$\mathrm{Re}\left[Y_J(\omega;\varphi)\right] = \frac{G_N}{2\hbar\omega}\int d\xi_L\int d\xi_R\frac{1}{2}\left(1 + \frac{\xi_L}{\epsilon_L}\frac{\xi_R}{\epsilon_R} + \frac{|\Delta_L|}{\epsilon_L}\frac{|\Delta_R|}{\epsilon_R}\cos\varphi\right)$$
$$\times\{f(\epsilon_R)[1 - f(\epsilon_L)][\delta(\epsilon_L - \epsilon_R - \hbar\omega) - \delta(\epsilon_L - \epsilon_R + \hbar\omega)] + (L\leftrightarrow R)\}. \tag{56}$$

The quasiparticle distribution functions here do not have to be equilibrium ones; they merely represent occupation factors of various energy states. An assumption of $L/R$ symmetry allows us to simplify Eq. (56):

$$\mathrm{Re}\left[Y_J(\omega;\varphi)\right] = \frac{2G_N}{\hbar\omega}\int_\Delta d\epsilon\frac{\epsilon(\epsilon + \hbar\omega) + \Delta^2\cos\varphi}{\sqrt{(\epsilon + \hbar\omega)^2 - \Delta^2}\sqrt{\epsilon^2 - \Delta^2}} \tag{57}$$
$$\times\left[f(\epsilon)(1 - f(\epsilon + \hbar\omega)) - f(\epsilon + \hbar\omega)(1 - f(\epsilon))\right].$$

The $\varphi$-dependence here comes from the interference between two processes of charge-$e$ transfer across the barrier: the first one consists of forwarding an electron as a quasiparticle across the barrier; the second process involves forwarding a Cooper pair accompanied by returning an electron. The involvement of the condensate makes the result of interference phase-dependent; the closer the quasiparticle energy $\epsilon$ to the gap, the stronger the relative effect of interference, cf. the numerator of the integrand in Eq. (57).

Now assume that the microwave energy quantum $\hbar\omega$ and the characteristic value $T_{\mathrm{eff}}$ of quasiparticle energy measured from the gap edge, $\epsilon - \Delta$, are small, $\hbar\omega, T_{\mathrm{eff}} \ll \Delta$. Then, by changing the variable of integration in Eq. (57) via $\epsilon = \Delta(1 + x)$ we find

$$\mathrm{Re}\left[Y_J(\omega;\varphi)\right] = \frac{1 + \cos\varphi}{2}\mathrm{Re}\left[Y_{\mathrm{qp}}(\omega)\right]. \tag{58}$$

Under an additional assumption $T_{\mathrm{eff}} \ll \hbar\omega$, the quasiparticle admittance here is

$$\mathrm{Re}\left[Y_{\mathrm{qp}}(\omega)\right] = \frac{1}{2}G_N\left(\frac{2\Delta}{\hbar\omega}\right)^{3/2}x_{\mathrm{qp}}. \tag{59}$$

The dimensionless quasiparticle density

$$x_{\mathrm{qp}} = \sqrt{\pi}\int_0^\infty dx\frac{f(\Delta(1 + x))}{\sqrt{x}} \tag{60}$$

is introduced here consistently with Eq. (25); it represents the density of quasiparticles $n_{\mathrm{qp}} = n_{\mathrm{CP}}x_{\mathrm{qp}}$ normalized by the density of Cooper pairs $n_{\mathrm{CP}} = 2\nu_0\Delta$ and assumes symmetric junction ($\nu_0 = \nu_L = \nu_R$).

The factor $x_{qp}$ in Eq. (59) accounts for the fact that all low-energy quasiparticles can absorb energy and hence contribute to dissipation. In equilibrium, $x_{qp}$ is controlled solely by the ratio $T/\Delta$, as the chemical potential $\mu$ of quasiparticles is pinned to zero. The simplest model of a non-equilibrium distribution allows for some $\mu \neq 0$ and effective temperature $T_{eff}$, and treats $x_{qp}$ and $T_{eff}$ as independent parameters. Leaving a more detailed discussion of the quasiparticle kinetics for Section 5, we mention here that there are reasons to expect $T_{eff} = T$, as a single quasiparticle may relax its energy by emitting a phonon. On the contrary, in order to recombine that quasiparticle has to meet another existing quasiparticle. Therefore, the recombination rate is smaller by a factor $x_{qp} \ll 1$ than the rate of quasiparticle energy relaxation.

Assuming the quasiparticle distribution is described by a Fermi function with some $\mu$ and effective temperature $T_{eff}$, the imaginary part of the quasiparticle admittance $Y_{qp}$ can be recovered using Kramers-Krönig transform. This would account for the effect of itinerant quasiparticles, but miss the main contribution to the imaginary part of the junction admittance $Y_J$, which originates from the response of the condensate (or, equivalently, from the contribution of Andreev bound states). We refer to Ref. [10] for a discussion of these points. In the next section we will relate the transition rates in superconducting qubits to the admittance of the Josephson junction.

## 3 Qubit transitions driven by quasiparticles

### 3.1 Qubit interaction with quasiparticles

For a Josephson junction shunted by an inductive loop, Fig. 2, the low-energy effective Hamiltonian can be written as

$$\mathcal{H} = \mathcal{H}_{\varphi} + \mathcal{H}_{qp} + \mathcal{H}_T^{qp}. \tag{61}$$

The first term determines the dynamics of the phase degree of freedom in the absence of quasiparticles, see Eq. (31). The contribution from pair tunneling, Eq. (45), is taken into account by the $E_J$ term in Eq. (31).

The second term in Eq. (61) is the sum of the BCS Hamiltonians for quasiparticles in the leads

$$\mathcal{H}_{qp} = \sum_{\alpha=L,R} \mathcal{H}_{qp}^{\alpha}, \tag{62}$$

with $\mathcal{H}_{qp}^{\alpha}$ of Eq. (14). The last term in Eq. (61) is the single quasiparticle tunneling Hamiltonian defined in Eq. (44) with a caveat: now the phase entering in $\mathcal{H}_T^{qp}$ is an operator,

$$\hat{H}_T^{qp} = \sum_{n_L,n_R,\sigma} t_{n_L n_R} \left( \left| u_{n_L} u_{n_R}^* \right| e^{i\hat{\varphi}/2} - \left| v_{n_L} v_{n_R}^* \right| e^{-i\hat{\varphi}/2} \right) \gamma_{n_L \sigma}^{\dagger} \gamma_{n_R \sigma} + \text{h.c.} \tag{63}$$

This way, $\mathcal{H}_T^{qp}$ becomes a Hamiltonian of the quasiparticles-qubit interaction. This way of accounting for the interaction is fine, as long as the dynamics of the condensate involves frequencies $\omega/2\pi$ much smaller that $\Delta/\pi\hbar$. Fortunately, this is the case for the typical devices controlled by microwaves (with frequency $\lesssim 10\,\text{GHz}$, while $\Delta/\pi\hbar \sim 100\,\text{GHz}$ in Al).

Within the described model, we can calculate the transition rate $\Gamma_{if}$ between qubit states $|i\rangle$ and $|f\rangle$ (i.e., eigenstates of the Hamiltonian $\mathcal{H}_{\varphi}$) associated with tunneling of a quasiparticle across the junction similarly to the calculation of admittance in Sec. 2.4. That is, we treat $\mathcal{H}_T^{qp}$ as a perturbation and evaluate the transition rates using Fermi's Golden Rule:

$$\Gamma_{if} = \frac{2\pi}{\hbar} \sum_{\lambda} \langle\!\langle \left| \langle f, \lambda | \mathcal{H}_T^{qp} | i, \eta \rangle \right|^2 \delta\left( E_{\lambda} - E_{\eta} - \hbar\omega_{if} \right) \rangle\!\rangle_{qp}, \tag{64}$$

where $\hbar\omega_{if} = E_i - E_f$ is the difference between the energies of the two qubit states. After averaging over initial quasiparticle states $|\eta\rangle$ and summing over final quasiparticle states $|\lambda\rangle$, and assuming $|\Delta_L| = |\Delta_R| \equiv \Delta$, we find

$$\Gamma_{if} = \frac{16E_J}{\hbar\pi\Delta} \int_\Delta d\epsilon\, f(\epsilon)\big[1 - f(\epsilon + \hbar\omega_{if})\big] \tag{65}$$

$$\Bigg[\frac{\epsilon(\epsilon + \hbar\omega_{if}) + \Delta^2}{\sqrt{\epsilon^2 - \Delta^2}\sqrt{(\epsilon + \hbar\omega_{if})^2 - \Delta^2}} \bigg|\langle f|\sin\frac{\varphi}{2}|i\rangle\bigg|^2$$

$$+ \frac{\epsilon(\epsilon + \hbar\omega_{if}) - \Delta^2}{\sqrt{\epsilon^2 - \Delta^2}\sqrt{(\epsilon + \hbar\omega_{if})^2 - \Delta^2}} \bigg|\langle f|\cos\frac{\varphi}{2}|i\rangle\bigg|^2 \Bigg].$$

It is important to note that the transitions between the qubit states are accompanied by the charge-$e$ transfer across the junction. In some cases, see Section 3.5, that helps one to single out the qubit transitions driven by quasiparticles.

For generic states and flux bias, the matrix elements of $\sin(\varphi/2)$ and $\cos(\varphi/2)$ have similar orders of magnitude; then, at low effective temperature of quasiparticles ($T_{\text{eff}} \ll \Delta$) the $\cos\varphi/2$ contribution to Eq. (65) is suppressed by a small factor $\sim \hbar\omega_{if}/\Delta$. Important exceptions are (quasi)elastic transitions (see Sec. 3.5) and transitions at special values of flux bias fine-tuned to suppress the $\sin\varphi/2$ matrix element (see Sec. 3.4).

## 3.2 Qubit energy relaxation

Here we focus on the relaxation rate from the first excited, $|i\rangle = |1\rangle$, to the ground, $|f\rangle = |0\rangle$, state in a generic setting, therefore neglecting the $\cos\varphi/2$ contribution, see the last line in Eq. (65).

With the assumptions discussed above, the qubit relaxation rate due to quasiparticle tunneling can be expressed as

$$\Gamma_{10} = \bigg|\langle 0|\sin\frac{\varphi}{2}|1\rangle\bigg|^2 S_{\text{qp}}(\omega_{10}), \tag{66}$$

where

$$S_{\text{qp}}(\omega) = \frac{16E_J}{\hbar\pi\Delta} \int_\Delta d\epsilon\, f(\epsilon)\big[1 - f(\epsilon + \hbar\omega)\big] \frac{\epsilon(\epsilon + \hbar\omega) + \Delta^2}{\sqrt{\epsilon^2 - \Delta^2}\sqrt{(\epsilon + \hbar\omega)^2 - \Delta^2}} \tag{67}$$

is the quasiparticle current spectral density, see the first two lines in Eq. (65). The quasiparticle states occupation factors $f(\epsilon)$ typically are small for all allowed energies, whether quasiparticles are at equilibrium or not, $f(\epsilon) \ll 1$. This allows us to replace $1 - f(\epsilon + \hbar\omega) \to 1$ in Eq. (67). Then, at low effective temperatures and frequencies, $T_{\text{eff}}, \hbar\omega \ll \Delta$, the quasiparticle current spectral density takes the form

$$S_{\text{qp}}(\omega) = \frac{8E_J}{\hbar\pi} x_{\text{qp}}\sqrt{\frac{2\Delta}{\hbar\omega}}, \quad \omega > 0. \tag{68}$$

By comparing this formula to Eq. (59), we find the relation

$$S_{\text{qp}}(\omega) = \frac{\omega}{\pi}\frac{1}{G_Q}\text{Re}\,Y_{\text{qp}}(\omega). \tag{69}$$

For a quasi-equilibrium distribution of quasiparticles with some $x_{\text{qp}}$ and $T_{\text{eff}}$, it may be viewed as a particular case of more general fluctuation-dissipation relations [10]. That allows one to conclude that

$$\Gamma_{01} = \Gamma_{10} \cdot e^{-\hbar\omega_{10}/T_{\text{eff}}} \tag{70}$$

indicating that $\Gamma_{10} > \Gamma_{01}$, as $T_{\text{eff}} > 0$ (there are no reasons to expect an inversion in the energy distribution of quasiparticles).

Relations (66) and (68) may allow one to link the magnitude of the junction dissipation (the real part of $Y_{\text{qp}}$) to the qubit relaxation rate. In the next sections we explore similarities and differences between the phase dependence of the admittance, see Eq. (58), and the variation of the relaxation rate with the external flux which controls the qubit and implicitly enters in Eq. (66).

### 3.3 Energy relaxation of a weakly anharmonic qubit

The last two terms (the "potential energy") in Eq. (31) in general possesses multiple minima, whose positions $\varphi_0$ are solutions of

$$E_J \sin \varphi_0 + E_L (\varphi_0 - 2\pi\Phi_e/\Phi_0) = 0 \,. \tag{71}$$

So long as the external flux is tuned away from half-integer multiples of the flux quantum and phase fluctuations are small, we can treat the potential energy in the harmonic approximation,

$$\mathcal{H}_\varphi \approx \mathcal{H}_\varphi^{(2)} = 4E_C N^2 + \frac{1}{2}(E_L + E_J \cos \varphi_0)(\varphi - \varphi_0)^2 \,. \tag{72}$$

The assumption of small phase fluctuations corresponds to the condition

$$\frac{E_C}{\hbar\omega_{10}} \ll 1 \,, \tag{73}$$

where

$$\omega_{10} = \sqrt{8E_C(E_L + E_J \cos \varphi_0)}/\hbar \tag{74}$$

is the qubit frequency in the harmonic approximation. For the transmon [6], $E_L = 0$ and $\varphi_0 = 0$, the condition (73) corresponds to the requirement of a large ratio between Josephson and charging energy, $E_J/E_C \gg 1$, which also enables us to neglect the dimensionless voltage $n_g$ of Eq. (31).

Within the harmonic approximation, it is straightforward to calculate the matrix element in Eq. (66) by expanding $\sin \varphi/2$ to linear order around $\varphi_0$. This way we find

$$\left| \langle 0 | \sin \frac{\varphi}{2} | 1 \rangle \right|^2 = \frac{E_C}{\hbar\omega_{10}} \frac{1 + \cos \varphi_0}{2} \,. \tag{75}$$

Substituting this expression into Eq. (66), and using Eq. (69) and the definition of the charging energy, we arrive at

$$\Gamma_{10} = \frac{1}{C} \text{Re}\, Y_{\text{qp}}(\omega_{10}) \frac{1 + \cos \varphi_0}{2} \,. \tag{76}$$

Therefore, the qubit relaxation rate is given by the inverse of the classical RC time of the junction, where $1/R$ is identified with the real part of its flux-dependent admittance [$\varphi_0$ is identified with the phase difference $\varphi$ in Eq. (58)]. This result seems to indicate that, as it is often the case, the behavior of the quantum harmonic oscillator is analogous to that of its classical counterpart. However, the analogy has its limitations, set by the form of the perturbation causing the relaxation: the selection rules for matrix elements of an operator $\sin \hat{\varphi}/2$ are less restrictive than for $\hat{\varphi}$. That opens a possibility, *e.g.*, of a direct decay from the second level to the ground state; the corresponding rate is [10]

$$\Gamma_{20} = \frac{1}{C} \text{Re}\, Y_{\text{qp}}(2\omega_{10}) \frac{1 - \cos \varphi_0}{2} \frac{E_C}{\hbar\omega_{10}} \,. \tag{77}$$

The dependence on phase/flux in this expression clearly differs from that in Eq. (58). Moreover, we remind that Eq. (76) is restricted to fluxes away from half-integer multiples of the flux quantum, so the relation to Eq. (58) does not necessarily hold at arbitrary flux – we explore this issue further in the next section.

### 3.4 The $\cos\varphi$ problem

The result for the real part of the junction admittance, Eq. (58), shows that as the phase difference approaches $\pi$, dissipation is suppressed. This is a manifestation of quantum mechanical interference: a quasiparticle is a coherent superposition of electron and hole-like excitations [cf. Eq. (10)], and these two components interfere during a tunneling event in a way that can preclude the absorption of energy. The exact cancellation at $\varphi = \pi$ is expected only in the limit of small temperature; more generally, one can write

$$\mathrm{Re}\left[Y_J(\omega;\varphi)\right] = \frac{1 + \varepsilon\cos\varphi}{2}\mathrm{Re}\left[Y_{\mathrm{qp}}(\omega)\right], \tag{78}$$

where $\varepsilon \to 1$ as $T \to 0$, see [15] and Ch. 2.6 in Ref. [12]. In the latter reference, experimental attempts to determine $\varepsilon$ in the 1970s are summarized, and the discrepancy between theory and experiments was termed "the $\cos\varphi$ problem".

It is interesting to consider the behavior of the admittance when the junction is part of a loop, so that the flux $\Phi$ biases the junction, $\varphi = 2\pi f$ with $f = \Phi/\Phi_0$. Expanding Eq. (78) around $f = 1/2$ we find

$$\mathrm{Re}\left[Y_J(\omega;2\pi f)\right] \approx \left[\frac{1-\varepsilon}{2} + \varepsilon\pi^2\left(f - \frac{1}{2}\right)^2\right]\mathrm{Re}\left[Y_{\mathrm{qp}}(\omega)\right]. \tag{79}$$

A fluxonium qubit consists of a small junction shunted by an inductor (the latter can be either an array of junctions or a superconducting nanowire). The qubit transition frequency $\omega_{10}(f)$ depends on the external flux; moreover for this qubit, assuming $\varepsilon = 1$, one can show that the relaxation rate near $f = 1/2$ takes the form [15]

$$\Gamma_{10} = \frac{F^2}{4}\frac{\omega_{10}(1/2)}{\pi G_Q}\mathrm{Re}\left[Y_J(\omega_{10}(1/2);2\pi f)\right], \tag{80}$$

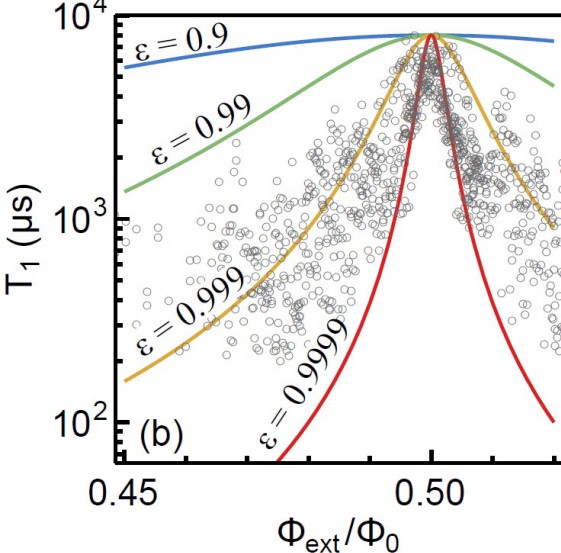

Figure 3: (from Ref. [15]) Empty circles: experimental data for $T_1$ at different values of external flux $\Phi_{\mathrm{ext}}$. Solid lines: theoretical rates calculated from Eq. (80) with $\mathrm{Re}[Y_J]$ of Eq. (79), for a few values of $\varepsilon$ and $F$ chosen to bound the data at $\Phi_{\mathrm{ext}}/\Phi_0 = 1/2$.

where the dimensionless prefactor $F$ can be calculated given the qubit parameters $E_C$, $E_J$, and $E_L$. With regard to the flux dependence, this expression extends the validity of Eq. (76) to the region near half flux quantum. Being a consequence of fluctuation-dissipation relations [cf. Eq. (69)], Eq. (80) can be expected to hold also for $\varepsilon \neq 1$. Therefore, measuring the flux dependence of the relaxation rate makes it possible to estimate the value of $\varepsilon$. The result of the measurements together with theoretical curves for different values of $\varepsilon$ are shown in Fig. 3; conservatively, one can estimate $\varepsilon > 0.99$.

### 3.5 Quasiparticles-driven $e$-jumps in a transmon

A qubit without an inductor is described by Hamiltonian (31) with $E_L = 0$ acting in a space of $2\pi$-periodic wave functions. If the ratio $E_J/E_C$ is not too large (typically, less than about 25 for a transmon), the dependence of the energy levels on $n_g$ is well-resolved in experiments. As we already discussed in Sec. 2.1, $n_g$ exhibits uncontrollable jumps by $\pm 1/2$ associated with the quasiparticle tunneling ($e$-jumps). Therefore, at a given $\overline{n}_g$ a transmon is not a two-level system, but in fact is a four-level system: two states differing by the charge parity represent the logical ground state, and another pair forms the logical excited state. Quasiparticle tunneling results in the parity-changing transitions within the four levels. The relaxation rates between the logical states considered in the previous section are one example of the $e$-jumps, but transitions which do not change the qubit logical state while changing its parity are also possible, see Fig. 4. In our notations, see Eq. (65), these rates are $\Gamma_{00}$ and $\Gamma_{11}$, for the transitions within the logical ground and excited states, respectively; they are also known as parity switching rates.

Rates $\Gamma_{10}$ and $\Gamma_{01}$ contribute to the energy relaxation rate $1/T_1$ of the qubit. The presence of finite rates $\Gamma_{00}$ and $\Gamma_{11}$ contribute to the qubit dephasing. The dephasing manifestation depends on the type of experiment. Suppose first the phase evolution of a qubit may be measured over time intervals shorter than $1/(\Gamma_{00}+\Gamma_{11})$, followed by averaging over many measurements. The energy levels of the qubit $E_1$ and $E_0$ would fluctuate from one measurement to another due to the $e$-jumps occurring between the measurements (cf. Section 2.1) and therefore the

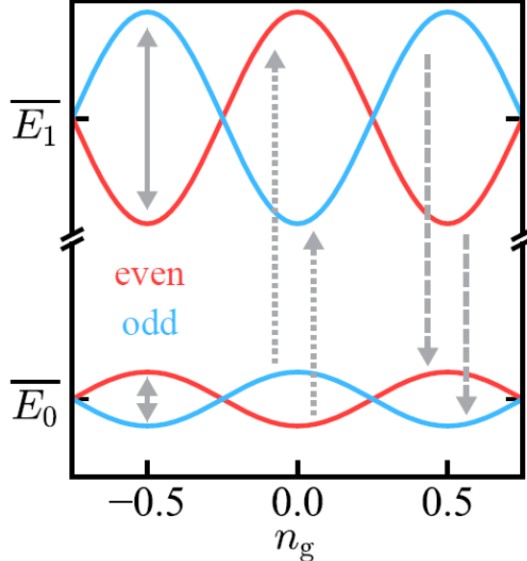

Figure 4: (adapted from Ref. [19]) Two lowest energy levels of the transmon qubit. The "even" and "odd" labels mark states of opposite charge parity.

frequency $\omega_{10}$ would fluctuate by some $\delta\omega = (\delta E_1 + \delta E_0)/\hbar$. The *averaged over the measurements result* then would yield the phase relaxation time $T_2^\star \sim 1/\delta\omega$, in analogy with the inhomogeneous broadening [16] of magnetic resonance (assuming we identify $\delta\omega$ with the spread of the magnetic resonance frequencies caused by static disorder in a solid). In a further analogy with the solid-state magnetic resonance, this relaxation mechanism is successfully countered by the echo technique [17], if the time $1/(\Gamma_{00} + \Gamma_{11})$ is long enough to allow for the echo pulse sequence. In the opposite case of high rates $\Gamma_{00}$ and $\Gamma_{11}$, fast e-jumps would lead to the phase diffusion with the diffusion constant $D_\varphi \sim (\delta E_1^2/\Gamma_{11} + \delta E_0^2/\Gamma_{00})/\hbar^2$.

In this Section, we evaluate all four rates, $\Gamma_{10}$, $\Gamma_{01}$, $\Gamma_{00}$, and $\Gamma_{11}$ for a transmon. We assume the quasiparticle distribution is described by an effective temperature $T_{\text{eff}} \ll \Delta$ and the dimensionless density $x_{\text{qp}} \ll 1$, which may or may not correspond to zero chemical potential.

Aiming at the most realistic case, we take $T_{\text{eff}} \ll \hbar\omega_{10}$, which allows us to use Eq. (68) in Eq. (66). Furthermore, disregarding small anharmonicity we utilize Eqs. (74) and (75) with $E_L = 0$ and $\varphi_0 = 0$, respectively, to find the qubit matrix element,

$$\left| \langle 0| \sin\frac{\varphi}{2} |1\rangle \right|^2 \approx \sqrt{\frac{E_C}{8E_J}}, \tag{81}$$

entering Eq. (66). As a result, the latter equation leads to

$$\Gamma_{10} = \frac{16E_J}{\hbar\pi} \sqrt{\frac{E_C}{8E_J}} \sqrt{\frac{\Delta}{2\hbar\omega_{10}}} x_{\text{qp}}. \tag{82}$$

Generalization to arbitrary $T_{\text{eff}}/\hbar\omega_{10}$ amounts to multiplication of the right-hand side of Eq. (82) by $\sqrt{4\hbar\omega_{10}/\pi T_{\text{eff}}}\exp(\hbar\omega_{10}/2T_{\text{eff}})K_0(\hbar\omega_{10}/2T_{\text{eff}})$, where $K_0(z)$ is a modified Bessel function, see [18]. Rate $\Gamma_{01}$ can be obtained from Eqs. (82) and (70).

Within the harmonic approximation, the level-preserving transitions ($0 \to 0$ and $1 \to 1$) occur between states which, up to exponentially small corrections [10], have the same parity of the wave function $\psi(\varphi)$. Therefore, rates $\Gamma_{00}$ and $\Gamma_{11}$ come only from the $\langle i| \cos(\varphi/2) |i\rangle \approx 1$ term in Eq. (65). A straightforward evaluation [18] yields

$$\Gamma_{00} \approx \Gamma_{11} \approx \frac{16E_J}{\hbar\pi} \sqrt{\frac{T_{\text{eff}}}{2\pi\Delta}} x_{\text{qp}}. \tag{83}$$

In the derivation of these rates we accounted for the near-degeneracy between the states connected by the transitions. Indeed, the corresponding energy difference (divided by $\hbar$) is at most a few MHz, so it satisfies the condition $\hbar\omega \ll T_{\text{eff}}$, even is we assume that quasiparticles equilibrate at the fridge temperature (10 mK $\approx$ 200 MHz).

Comparing Eqs. (82) and (83), we find the ratio of the rates, $\Gamma_{11}/\Gamma_{10} = (8E_J/E_C)^{1/2} \cdot (\hbar\omega_{10}T_{\text{eff}}/\pi\Delta^2)^{1/2}$. The first factor here is large, while the second one is small. For parameters of a typical transmon, the second factor wins the competition, so that $\Gamma_{11}/\Gamma_{10} \ll 1$; therefore it is unlikely for e-jumps to cause any phase diffusion.

# 4 Photon-assisted e-jumps

## 4.1 Generation of quasiparticles by photons

So far we have considered transitions in a qubit due to quasiparticles, but neglected any effect of the external environment. In general, a qubit is coupled to the environment via a cavity or a waveguide resonator which support a number of modes; in other words, the qubit is coupled to photons whose frequencies $\omega_\nu$ depend on the geometry of the device. We can distinguish two

kinds of photons: those with frequency lower than the pair-breaking energy, $\omega_\nu < 2\Delta/\hbar$, can be absorbed or emitted during a tunneling event of a quasiparticle already present, similarly to the absorption/emission of the qubit energy by the quasiparticles. Assuming that only a small number of such low-frequency photons are present in the cavity, this type of photon-assisted tunneling can only contribute a small correction (which we neglect) to the rates calculated in the previous section. In contrast, higher frequency photons, $\omega_\nu > 2\Delta/\hbar$, can always be absorbed, even in the absence of quasiparticles, as they have enough energy to break a Cooper pair and thus create two quasiparticles. The photon wave length, even at the photon energy somewhat exceeding $2\Delta$, is still comparable to the size of the qubit. Therefore it is fair to assume that the alternating voltage generated by a photon is applied across the junction. Thus breaking of a Cooper pair generates two quasiparticles, one on each side of the junction. Here we consider the $e$-jump rates associated with such pair-breaking events.

### 4.2 Theory of photon-assisted $e$-jumps

The inclusion of the qubit-photon interaction in our model can be accomplished [20] by adding to Eq. (61) the Hamiltonian for the photon (we focus on one mode of a cavity here for simplicity)

$$\mathcal{H}_{\text{cav}} = \hbar\omega_\nu b_\nu^\dagger b_\nu, \tag{84}$$

where $b_\nu^\dagger$ and $b_\nu$ are the creation and annihilation operators for the photon, and by replacing

$$\varphi \to \varphi + \phi_\nu\left(b_\nu + b_\nu^\dagger\right) \tag{85}$$

in Eqs. (46). Here $\phi_\nu$ is the amplitude of zero-point fluctuation of the phase due to the electric field $\mathcal{E}_\nu(0)$ at the junction position:

$$\phi_\nu = \frac{2e d_\nu \mathcal{E}_\nu(0)}{\hbar\omega_\nu}, \tag{86}$$

with $d_\nu$ being the effective dipole length that relates the electric field to the voltage drop $\mathcal{U}_\nu$ across the junction, $\mathcal{U}_\nu = d_\nu\mathcal{E}_\nu(0)$. With these definitions, one can recognize that the replacement in Eq. (85) originates from the relation between phase and voltage in Eq. (36), see also the text preceding Eq. (53).

For our purposes, it is sufficient to perform the replacement (85) in Eq. (45) and then consider the first term in the expansion over the small parameter $\phi_\nu \ll 1$. In this way, we obtain the following quasisparticle-qubit-photon interaction term:

$$\delta\mathcal{H}_T = \frac{i\phi_\nu}{2}\left(b_\nu + b_\nu^\dagger\right)\sum_{n_L n_R \sigma}\sigma t_{n_L n_R}\left(\left|u_{n_L} v_{n_R}\right|e^{i\frac{\varphi}{2}} - \left|v_{n_L} u_{n_R}\right|e^{-i\frac{\varphi}{2}}\right)\gamma_{n_L\sigma}^\dagger \gamma_{n_R\bar{\sigma}}^\dagger$$
$$+ \text{h.c.} \tag{87}$$

Using as before Fermi's golden rule, we can find the transition rate $\Gamma_{if}^{\text{ph}}$ between the initial state with qubit in state $|i\rangle$, no quasiparticles, and one photon in the cavity, and the final state with qubit in $|f\rangle$, two quasiparticles, and no photon:

$$\Gamma_{if}^{\text{ph}} = \Gamma_\nu\left[\left|\langle f|\cos\frac{\varphi}{2}|i\rangle\right|^2 S_-\left(\frac{\hbar\omega_\nu + \hbar\omega_{if}}{\Delta}\right) + \left|\langle f|\sin\frac{\varphi}{2}|i\rangle\right|^2 S_+\left(\frac{\hbar\omega_\nu + \hbar\omega_{if}}{\Delta}\right)\right], \tag{88}$$

where

$$\Gamma_\nu = \frac{2}{\hbar\pi}\phi_\nu^2 E_J \tag{89}$$

can be related to the coupling strength between qubit and cavity [20], and

$$
\begin{aligned}
S_\pm(x) &= \int_1 dy \int_1 dz \, \frac{yz \pm 1}{\sqrt{y^2-1}\sqrt{z^2-1}} \delta(x-y-z) \\
&= (x+2)E\left(\frac{x-2}{x+2}\right) - 4\frac{x+1\mp1}{x+2}K\left(\frac{x-2}{x+2}\right),
\end{aligned}
\tag{90}
$$

with $E$ and $K$ the complete elliptic integrals of the second and first kind, respectively. These structure factors have the following properties: $S_\pm(x) = 0$ for $x < 2$, $S_\pm(x) \approx x$ for $x \gg 2$, and

$$
S_+(x) \approx \pi[1 + (x-2)/4]
\tag{91}
$$

$$
S_-(x) \approx \pi(x-2)/2
\tag{92}
$$

for $x - 2 \ll 2$. Note the similarities between Eqs. (65) and (88): in both there are a prefactor that accounts for the coupling strength, and the squared matrix element of $\sin\varphi/2$ and $\cos\varphi/2$ multiplied by qubit-frequency-dependent structure factors. The latter additionally depend on the quasiparticle distribution function in Eq. (65) or on the photon frequency in Eq. (88); this difference relates to the different physical origin of the transitions: those with rates in Eq. (65) require quasiparticles to be present but no photons, while those in Eq. (88) require the presence of photons but not quasiparticles.

Let us consider again the case of a single junction transmon; then using Eq. (88) we find for the parity switching, relaxation, and excitation rates:

$$
\Gamma_{ii}^{\mathrm{ph}} \approx \Gamma_\nu S_-\left(\frac{\hbar\omega_\nu}{\Delta}\right)
\tag{93}
$$

$$
\Gamma_{10}^{\mathrm{ph}} \approx \Gamma_\nu \sqrt{\frac{E_C}{8E_J}} S_+\left(\frac{\hbar\omega_\nu + \hbar\omega_{10}}{\Delta}\right)
\tag{94}
$$

$$
\Gamma_{01}^{\mathrm{ph}} \approx \Gamma_\nu \sqrt{\frac{E_C}{8E_J}} S_+\left(\frac{\hbar\omega_\nu - \hbar\omega_{10}}{\Delta}\right).
\tag{95}
$$

The relaxation and excitation rates are generally close: for $\omega_{10} < \min\{\omega_\nu - 2\Delta/\hbar, 2\Delta/\hbar\}$ we find

$$
1 - \frac{\hbar\omega_{10}}{2\Delta} < \frac{\Gamma_{01}^{\mathrm{ph}}}{\Gamma_{10}^{\mathrm{ph}}} < 1,
\tag{96}
$$

with the lower bound saturated as $\omega_\nu \to 2\Delta/\hbar$ and the upper one for $\omega_\nu \to \infty$. This is in contrast to "cold" quasiparticles, in which case $\Gamma_{01}^{\mathrm{qp}}/\Gamma_{10}^{\mathrm{qp}} \ll 1$.

Finally, the ratio between parity switching and relaxation rates can be large, since

$$
\frac{\Gamma_{ii}^{\mathrm{ph}}}{\Gamma_{10}^{\mathrm{ph}}} \approx \sqrt{\frac{8E_J}{E_C}} \gg 1
\tag{97}
$$

for $\omega_\nu \gg 2\Delta/\hbar$. On the other hand, for photons near the pair-breaking threshold, $\hbar\omega_\nu - 2\Delta \ll \Delta$, we find

$$
\frac{\Gamma_{ii}^{\mathrm{ph}}}{\Gamma_{10}^{\mathrm{ph}}} \approx \sqrt{\frac{2E_J}{E_C}}\left(\frac{\hbar\omega_\nu}{\Delta} - 2\right)
\tag{98}
$$

and the large prefactor on the right hand side can be compensated by the small, final factor originating from $S_-(x)$.

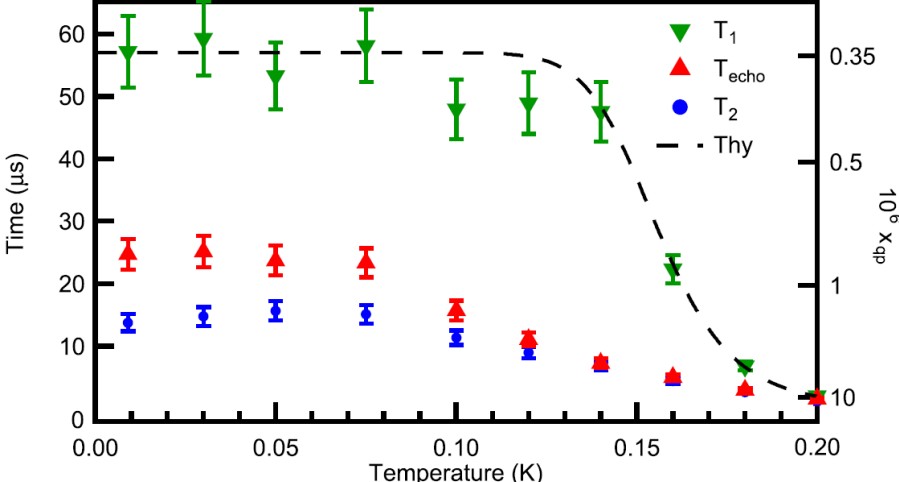

Figure 5: (adapted from Ref. [21]) Symbols with error bars: experimental data for $T_1$, $T_2$, and $T_{\text{echo}}$ vs temperature. Dashed line: theoretical $T_1$ time calculated using Eq. (99).

## 4.3 Comparison with experiments

Various experiments have reported measurement of qubit rates for transitions that can be caused by quasiparticles. In the work introducing the 3D transmon architecture [21], the $T_1$ time was measured as function of temperature $T$. At low temperatures, this time was roughly independent of $T$, while it became quickly shorter at higher temperature, see Fig. 5. A possible explanation of this behavior is that at low temperatures there are non-equilibrium, cold quasiparticles with the density $x_{\text{qp}} \approx 3 \cdot 10^{-7}$. This value is exceeded by the density $x_{\text{qp}}^{eq}$ of equilibrium thermally-activated quasiparticles at $T \gtrsim 120\text{mK}$. Consistently with that, the relaxation time drops quickly upon the further increase of temperature. The quasiparticle-driven relaxation rate scales linearly with the quasiparticle density, so we may separate the contributions of $x_{\text{qp}}$ and $x_{\text{qp}}^{eq}$ to $1/T_1$,

$$\frac{1}{T_1} = \frac{1}{T_1^0} + \frac{1}{T_1^{eq}} \, . \tag{99}$$

Alternatively, $1/T_1^0$ could be dominated by a different, non-quasiparticle mechanism. One may attempt to distinguish between the mechanisms by measuring $T_1$ as a function of flux [15] or attempting to separate the relaxation processes involving $e$-jumps from those preserving the charge parity [22]. The fluxonium experiment indicates the presence of quasiparticle-driven relaxation at low temperatures and yields $x_{\text{qp}} \approx (1 - 32) \cdot 10^{-7}$. In the transmon experiment [22] temperature-independent relaxation was dominated by mechanisms other than $e$-jumps.

A more recent experiment [19] in a similar setting (low $E_J/E_C$ ratio) extracted all the six transition rates between the four qubit states. We show in Fig. 6 the parity-changing, $e$-jump rates. Again we see that they are independent of temperature at low temperatures, and quickly increase at higher temperatures. Moreover, $\Gamma_{10}/\Gamma_{01} \sim 1$, giving a strong evidence for the photon-assisted $e$-jumps. In fact, except for $\Gamma_{11}$, we can fit the data assuming that the rates are given by the sums of the contributions stemming from thermal quasiparticles and photon-assisted $e$-jumps calculated in the previous two sections:

$$\Gamma_{ij} = \Gamma_{ij}^{\text{qp}} + \Gamma_{ij}^{\text{ph}} \, . \tag{100}$$

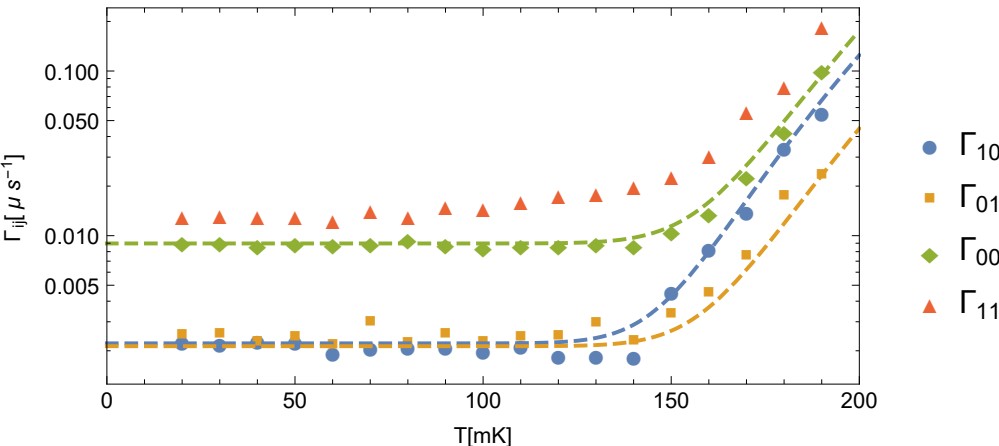

Figure 6: (data points from Ref. [19]) Symbols: experimental data for the transition rates vs temperature. Dashed lines: theoretical rates calculated using Eq. (100) and the parameters given in the text.

Some parameters ($E_C \simeq 355\,\mathrm{MHz}$, $E_J/E_C \simeq 22.8$, $\omega_{10}/2\pi = 4.400\,\mathrm{GHz}$) are obtained from independent measurements; then we are left with three fit parameters: $\Delta/2\pi \simeq 49.1\,\mathrm{GHz}$, $\hbar\omega_\nu/\Delta \simeq 2.8$, and $\Gamma_\nu \simeq 7.7\,\mathrm{kHz}$. The result of the fit is shown by the solid lines. The disagreement between theory [18, 20] predicting $\Gamma_{11}/\Gamma_{00} < 1$ and experiment [19], showing $\Gamma_{11}/\Gamma_{00} \approx 1.3$ is currently unexplained.

# 5 Quasiparticle dynamics

## 5.1 Energy relaxation, recombination, and trapping

While multiple experiments indicate a low-temperature saturation of the quasiparticle density at a level $x_{\mathrm{qp}} \sim 10^{-7} - 10^{-6}$, the source of such non-equilibrium population is not uniquely identified. Monitoring of the occupation of the fluxonium states [23] over a $\sim 10$ min time span indicates that quasiparticles arrive in bunches, which leads to a non-Poissonian statistics of the quantum jumps between the qubit states. The qubit temperature (measured by the relative occupation of the states $|0\rangle$ and $|1\rangle$) remains low, favoring the assumption that the non-equilibrium quasiparticles are also cold, $T_{\mathrm{eff}} \approx 40 - 60$ mK, see Figs. 7(a) and 7(b).

Quasiparticles may relax their energy in a superconductor by emitting phonons. Recombination is also accompanied by emission of a phonon that carries away the energy ($\sim 2\Delta$) released in the annihilation of two quasiparticles (in this discussion, we focus on the zero-temperature limit and low quasiparticle densities). Recombination requires a meeting of two quasiparticles, therefore the corresponding rate equation has the form $dx_{\mathrm{qp}}/dt \propto -x_{\mathrm{qp}}^2$. The recombination rate per quasiparticle scales as $1/\tau_r \propto x_{\mathrm{qp}}$. The relaxation rate, on the other hand, is independent of $x_{\mathrm{qp}}$ but has a strong dependence on the energy $E$ of the quasiparticle measured from the gap. The electron-phonon relaxation rate in metals is strongly affected by disorder. For thin films, the "dirty limit" in which the superconducting coherence length exceeds the electron elastic mean free path, is an adequate approximation. The theory of these rates is beyond the scope of these lectures; its summary can be found in Ref. [24]. The same work provides a detailed theory of the relaxation ($1/\tau_E$) and recombination ($1/\tau_r$) rates for quasiparticles in superconductors (with or without applied magnetic field). Using the results

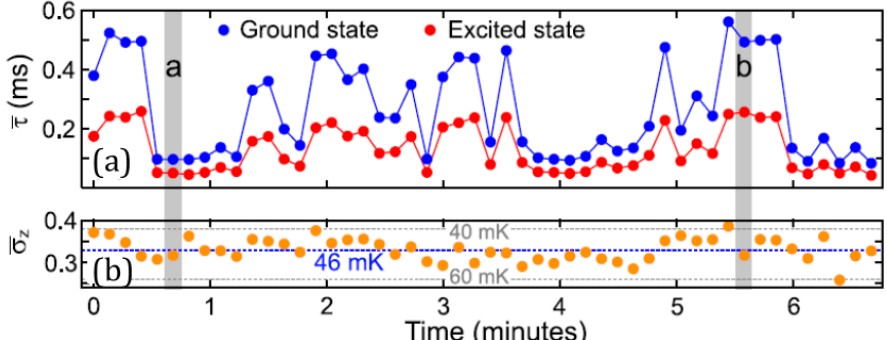

Figure 7: (adapted from Ref. [23]). (a) Measurement of the average time spent by the qubit in the ground (blue) and excited (red) states vs time. There are significant fluctuations in these values over the course of minutes. (b) Polarization of the flux-onium qubit vs time. The dashed blue line marks the average polarization, which corresponds to a temperature of 46 mK, and the gray dashed lines are markers for 40 and 60 mK. Note that the qubit temperature is not correlated with the fluctuations between the quiet and noisy intervals. The qubit polarization $\sigma_z$ is defined in analogy to a spin-1/2 polarization.

of [24] [specifically, Eqs. (57) at zero field, $T_{\text{eff}} \ll \Delta$, and $E \ll \Delta$] we find

$$\frac{1}{\tau_E} \approx \frac{3}{\tau_N(\Delta)} \left( \frac{E}{\Delta} \right)^{9/2}, \ \frac{1}{\tau_r} \approx \frac{1}{2\tau_N(\Delta)} \frac{T_{\text{eff}}}{\Delta} x_{\text{qp}}. \tag{101}$$

Here $1/\tau_N(\Delta)$ is the rate of relaxation of an electron with energy $\Delta$ (measured from the Fermi level) in the normal state. It depends, among other parameters on the electron mean free path. We extract the estimate $1/\tau_N(\Delta) = 4 \cdot 10^8 \, \text{s}^{-1}$ for Al with electron diffusion constant $D = 20 \, \text{cm}^2/\text{s}$ from the data of Ref. [25]. In finding the rates, we considered, respectively, relaxation of a quasiparticle by a spontaneous phonon emission and recombination of a quasiparticle with a background quasiparticle distribution characterized by $x_{\text{qp}}$ and $T_{\text{eff}}$. We also assumed that the thickness of the superconducting film exceeds the wavelength of a phonon with energy $E$; violation of that condition reduces [24] by 1 the exponent of the $E/\Delta$ factor in $1/\tau_E$.

Comparing the two rates of Eq. (101), we find that despite the precipitous drop of $1/\tau_E$ with energy, this rate still exceeds by an order of magnitude the recombination rate $1/\tau_r$ at $E = T_{\text{eff}} = 35$ mK and $x_{\text{qp}} = 10^{-6}$. The recombination rate may be further reduced by the re-absorption of phonons in the superconductor, recreating pairs of quasiparticles [26]. That provides one with the grounds for assuming for quasiparticles a Gibbs distribution with a finite chemical potential and $T_{\text{eff}} = T$. This assumption is further justified by the expectation that qubit manipulation with microwaves does not heat the quasiparticles [27].

The considered above processes may occur regardless the presence of macroscopic inhomogeneities in a superconductor. Inclusion of inhomogeneities, however, brings about two more elements of the quasiparticle dynamics, *i.e.*, their diffusion and trapping. Experiments with qubits have opened new ways to investigate these processes, based on monitoring the relaxation rate of a qubit. Indeed, an excess density of quasiparticles created by a AC bias jolt applied to the junction affects the qubit relaxation rate, see Eqs. (66) and (68). The total, time-dependent relaxation rate $\Gamma(t)$ of a transmon qubit can be written in the form

$$\Gamma(t) = \gamma x_{\text{qp}}(0, t) + \Gamma_0, \tag{102}$$

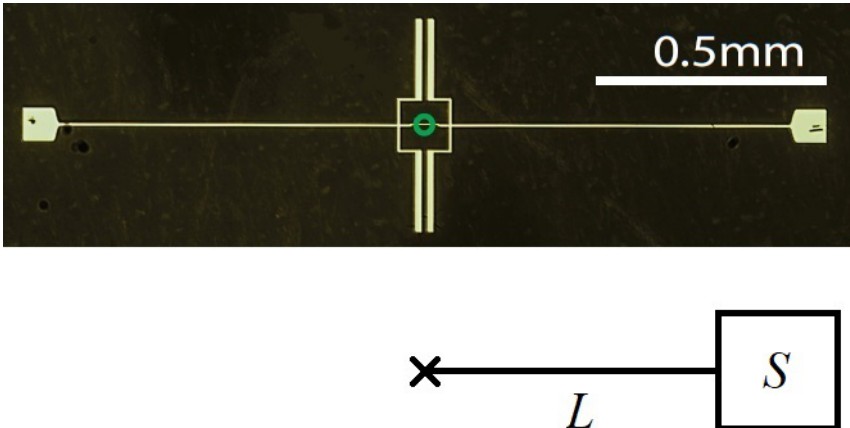

Figure 8: Top: (adapted from Ref. [30]) Optical image of a qubit used to measure quasiparticle trapping due to vortices. Bottom: schematic diagram of the simplified model used in the text to calculate the quasiparticle density decay rate; it represent one half of a symmetric device with a wire of length $L$ attached to a pad of area $S$.

where $x_{\mathrm{qp}}(0, t)$ is the time-dependent excess quasiparticle density at the position of the junction, while $\Gamma_0$ accounts for all other relaxation mechanisms (including relaxation due to a possibly finite steady-state quasiparticle density). Using Eq. (75) with $\varphi_0 = 0$ for the matrix element of a transmon, we have $\gamma \simeq \sqrt{2\Delta\omega_{10}/\hbar}/\pi$ for the constant in Eq. (102). In the next sections we consider first the effect of vortices, which hints at the possibility of affecting the dynamics and thus improve qubit performance. Motivated by these results, we then study the stronger effect of a normal-metal quasiparticle trap.

## 5.2 Single-vortex trapping power

When a thin superconducting film is cooled below its critical temperature in the presence of a perpendicular magnetic field, vortices are trapped in the film if the field is higher than a certain threshold. For a strip of width $W$, this threshold is of the order $\Phi_0/W^2$, see Ref. [28] for a detailed discussion (further discussion of a ring and disk geometries can be found in Ref. [29]). The threshold field is usually small, amounting to a few milliGauss for a strip or disk with a width of few tens of microns. This implies that vortices can be avoided or permitted in certain regions of a superconducting circuit by properly choosing the dimensions of its features. For example, for the qubit design in Fig. 8, top panel, at sufficiently low field the vortices will only be trapped into the large square pads at the ends of the long and thin antenna wire.

In the presence of a vortex, the superconducting order parameter is suppressed over a core region of radius $\sim \xi$, with $\xi$ the coherence length. In this region, a quasiparticle can loose energy (e.g., by emitting a phonon), since there are states available with energy below the bulk gap, and thus get trapped in the vortex core. Experiment [30] revealed the effect of a single vortex on the quasiparticle dynamics and measured the relevant quantity, the trapping power $P$ of a vortex, which is an intrinsic property of a vortex, independent of the device geometry.

At a phenomenological level, we expect the dynamics of the quasiparticle density to be

governed by the following generalized diffusion equation:

$$\frac{\partial x_{\text{qp}}(r,t)}{\partial t} = D_{\text{qp}}\nabla^2 x_{\text{qp}}(r,t) - P\sum_{i=1}^{N}\delta(r-r_i)x_{\text{qp}}(r,t), \tag{103}$$

where $D_{\text{qp}}$ is the (temperature-dependent) quasiparticle diffusion constant, $P$ is the "trapping power" of a single vortex, and the sum is over all the $N$ vortices at positions $R_i$. The trapping by vortices leads to an exponential decay of the density; we calculate this decay rate in a simplified model of a (quasi-)1D wire of length $L$ and width $W \ll L$ attached to a square pad of area $S$ (see bottom panel of Fig. 8). The diffusion equations in the wire and pad are, respectively:

$$\frac{\partial x_{\text{qp}}(y,t)}{\partial t} = D_{\text{qp}}\frac{\partial^2 x_{\text{qp}}(y,t)}{\partial y^2} \tag{104}$$

$$\frac{\partial x_{\text{qp}}(r,t)}{\partial t} = D_{\text{qp}}\nabla^2 x_{\text{qp}}(r,t) - P\sum_{i=1}^{N}\delta(r-r_i)x_{\text{qp}}(r,t), \tag{105}$$

with the boundary condition $\nabla_\perp x_{\text{qp}} = 0$ at the boundaries (here $\nabla_\perp$ denotes the gradient in the direction perpendicular to the boundary). While Eq. (104) can be easily solved analytically, this is not possible for Eq. (105). However, so long as the diffusion rate $D_{\text{qp}}/S$ inside the pad is fast compared to the density decay rate, the density within the pad can be taken to be approximately uniform and we can derive from Eq. (105) a boundary condition for Eq. (104) by integrating the former over the pad area to obtain

$$S\frac{\partial x_{\text{qp}}(L,t)}{\partial t} = -WD_{\text{qp}}\frac{\partial x_{\text{qp}}(y,t)}{\partial y}\bigg|_{y=L} - PNx_{\text{qp}}(L,t). \tag{106}$$

The solution to Eq. (104) can be written in the form:

$$x_{\text{qp}}(y,t) = e^{-st}\alpha\cos(ky), \tag{107}$$

where $s = D_{\text{qp}}k^2$ is the density decay rate (i.e., the total trapping rate), and the boundary condition at the origin is satisfied.[1] Then, using the boundary condition Eq. (106), we have the following equation for $k$:

$$z\tan z + \frac{S}{A_W}z^2 - N\frac{P\tau_D}{A_W} = 0, \tag{108}$$

with $z = kL$, $A_W = WL$ the wire area, and $\tau_D = L^2/D_{\text{qp}}$ the diffusion time along the wire. There are two distinct limits for the solution of Eq. (108).

In the limit of small number of vortices of weak trapping power, $NP\tau_D/A_W \ll 1$, we have $z^2 \approx NP\tau_D/A$, where $A = S + A_W$ is the total device area. In this regime, the density decay rate $s$ is then

$$s \approx \frac{NP}{A}, \tag{109}$$

which is proportional to the number of vortices. It is the vortices which provide the bottleneck for the quasiparticle evacuation from the vicinity of the Josephson junction. The measured in experiment rate $s$ exhibited step-wise increase with the external field, see Fig. 9. Each step corresponds to entering of a vortex in a pad. The step height allowed one to extract [30] the trapping power of a single vortex, $P \approx 0.067\,\text{cm}^2/\text{s}$.

---

[1]The vanishing of the density derivative at the junction position is equivalent to considering a symmetric device with the same number of vortices in both pads; generalizations to more complex geometry and unequal vortex number can be found in Ref. [30].

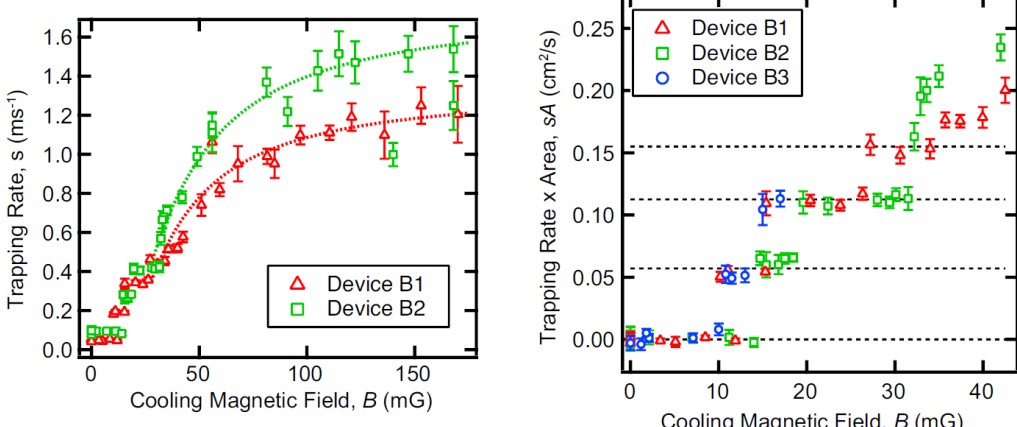

Figure 9: (adapted from Ref. [30]) Left: quasiparticle density decay rate (or trapping rate) vs cooling field for two devices . Right: trapping rate times area vs field for three devices; at low field, the stepwise increase of the rate is evident.

Comparing this finding with a theory remains to be a challenge. A crude estimate of the electron-electron interaction effect within the vortex core yields [30] trapping power smaller than the observed value by a factor $\sim 10^2$. The additional effect of the periphery of the vortex was considered in [24]. At the vortex periphery, the gap is suppressed compared to its nominal value; a propagating quasiparticle may emit a phonon and get trapped in that region. The additional rate associated with such process does not resolve discrepancy with the experiment, but indicates an interesting and yet unexplored temperature dependence of the trapping power.

As the number of vortices increases, the bottleneck shifts to the diffusion along the wires connecting the junction to the antenna pads. In terms of Eq. (108), it means the existence of an upper limit for the solution: $z \approx \pi/2$ for $NP\tau_D/A_W \gg 1$. In this case, the density decay rate is determined by the diffusion rate:

$$s \approx \frac{\pi^2}{4\tau_D}, \tag{110}$$

with diffusion coefficient $D_q \approx 20 \ \text{cm}^2/\text{s}$. Along with the stepwise increase of the decay rate for small vortex number, an upper bound for the decay rate was also measured, see Fig. 9.

## 5.3 Normal-metal traps

The core of a vortex can be thought of as a small (size $\xi^2$) normal-state region inside a superconductor. Since quasiparticles can be trapped there, one can expect that an actual normal-metal island can also act as a quasiparticle trap. Here we consider the case of such an island in tunnel contact with a superconductor – that is, the normal and superconducting layers are separated by a thin insulating barrier, so that the contact has low transparency. In this situation, we have again a generalized diffusion equation for the quasiparticle density:

$$\frac{\partial x_{\text{qp}}(r,t)}{\partial t} = D_{\text{qp}}\nabla^2 x_{\text{qp}}(r,t) - a(r)\Gamma_{\text{eff}}x_{\text{qp}}(r,t). \tag{111}$$

In the last term, the function $a(r)$ is unity in the normal-metal/superconductor contact region and zero elsewhere. The effective trapping rate $\Gamma_{\text{eff}}$ accounts for the competition of three effects (see Fig. 10): a quasiparticle in the superconductor can tunnel into the normal metal at

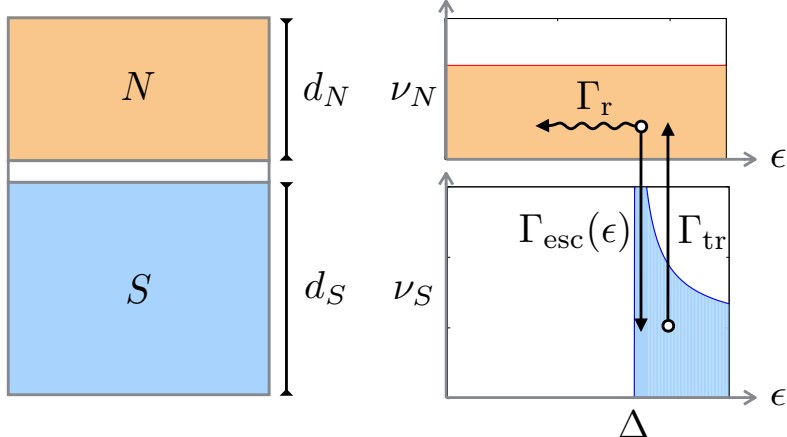

Figure 10: (from Ref. [31]) Left: a normal-metal layer $N$ of thickness $d_N$ is in tunnel contact with a superconductor $S$ of thickness $d_S$. Right: depiction of the processes determining the effective trapping rate: tunneling into the normal metal with rate $\Gamma_{\rm tr}$, relaxation to below the gap $\Delta$ with rate $\Gamma_{\rm r}$, and escape back into the superconductor with rate $\Gamma_{\rm esc}(\epsilon)$. While the normal-metal density of states is featureless (top), the superconductor's one is peaked at the gap and zero below it (bottom).

rate $\Gamma_{\rm tr}$; once in the normal metal, the excitation can relax to states with energy below the gap at rate $\Gamma_{\rm r}$, or it can escape back to the superconductor at rate $\Gamma_{\rm esc}(\epsilon)$. The latter is energy dependent because in calculating such a rate via Fermi's golden rule, the bare (energy-independent) tunneling-out rate $\Gamma_{\rm esc}$ is enhanced by singularity of the (normalized) BCS density of states,

$$\Gamma_{\rm esc}(\epsilon) = \Gamma_{\rm esc} \frac{\epsilon}{\sqrt{\epsilon^2 - \Delta^2}}, \quad \epsilon > \Delta. \tag{112}$$

Identifying the quasiparticle effective temperature with $T$, we can distinguish two limiting regimes (see Ref. [31] for details) for the effective trapping rate $\Gamma_{\rm eff}$: if relaxation is fast, $\Gamma_{\rm r} \gg \Gamma_{\rm esc}\sqrt{\Delta/T}$, then the "bottleneck" process is tunneling into the normal metal and $\Gamma_{\rm eff} \approx \Gamma_{\rm tr}$; if relaxation is slow, $\Gamma_{\rm r} \lesssim \Gamma_{\rm esc}\sqrt{\Delta/T}$, then relaxation is the bottleneck and

$$\Gamma_{\rm eff} \approx \frac{\Gamma_{\rm r}\Gamma_{\rm tr}}{\Gamma_{\rm esc}} \sqrt{T/\Delta}. \tag{113}$$

For both the slow and fast relaxation regimes, according to Eq. (111) the dynamics of the density is determined by diffusion and effective trapping rate. Similarly to the case of vortices, we can gain a qualitative understanding of the dynamics by studying a simplified model, see Fig. 11. Let us consider a superconducting strip of length $L + d$ and width $W \ll L$, with the region $-d < y < 0$ in contact with normal metal and the region $0 < y < L$ free; then Eq. (111) simplifies to

$$\partial_t x_{\rm qp}(y, t) = D_{\rm qp} \partial_y^2 x_{\rm qp}(y, t) - \theta(-y)\Gamma_{\rm eff} x_{\rm qp}(y, t), \tag{114}$$

with the boundary condition $\partial x_{\rm qp}/\partial y = 0$ at $y = -d, L$. So long as the trap is small, $d \ll \lambda_{\rm tr}$, compared with the trapping length defined as $\lambda_{\rm tr} = \sqrt{D_{\rm qp}/\Gamma_{\rm eff}}$, we can treat the trap in the same way as we treated the pad in the previous section, and from integrating over the trap area obtain an effective boundary condition at $y = 0$:

$$\partial_t x_{\rm qp}(0, t) = (D_{\rm qp}/d)\partial_y x_{\rm qp}(y, t)|_{y=0} - \Gamma_{\rm eff} x_{\rm qp}(0, t). \tag{115}$$

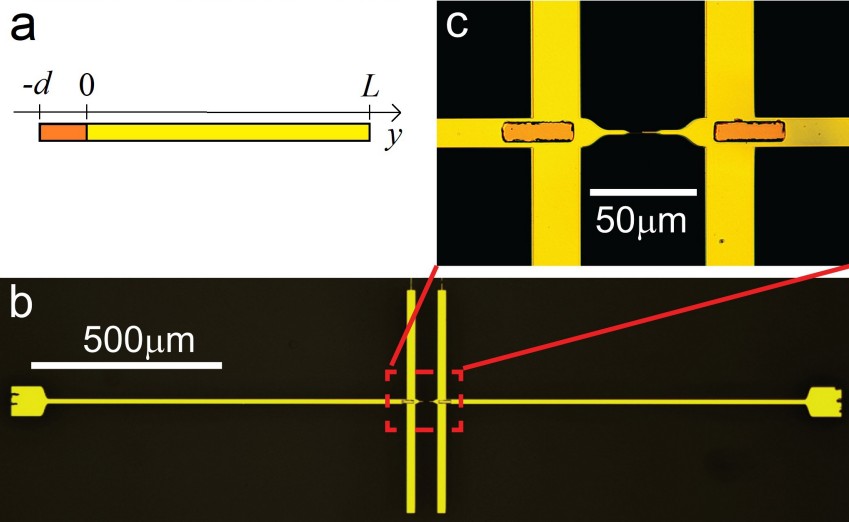

Figure 11: (adapted from Ref. [31]) a: simplified geometry used to study the dynamics of quasiparticle density in the presence of a normal-metal trap. The trap region (orange) goes from $-d$ to 0 and the superconductor without trap (yellow) from 0 to $L$. b: optical image of one of the devices used in the experiments. c: zoomed-in image of the copper traps (orange) deposited over aluminum (yellow) near the junction.

Taking the solution in the region $y > 0$ to be of the form $x_{qp} = e^{-st}\alpha\cos[k(y-L)]$ with $s = D_{qp}k^2$, and using this boundary condition, we arrive at

$$z\tan z + \frac{d}{L}z^2 - \frac{\pi}{2}\frac{d}{l_0} = 0,\qquad(116)$$

with $z = kL$ and $l_0 = \pi D_{qp}/2\Gamma_{eff}L = \pi\lambda_{tr}^2/2L$. Up to different parameters, this equation for $z$ has the same form as Eq. (108);[2] therefore, we find again two regimes, one for small, weak traps with the density decay rate given by

$$s \approx \frac{d}{d+L}\Gamma_{eff}\qquad(117)$$

valid for $d \ll l_0$, and one for large/strong traps ($d \gg l_0$) in which the decay rate is limited by diffusion, see eq. (110).

The cross-over between the weak and strong trapping regime can be studied by increasing the trap length $d$ while keeping everything else equal. Such experiments were carried out with transmon qubits of design similar to that used in the vortex experiments, see Fig. 11. In Fig. 12 we show the (normalized) density decay rate vs (normalized) trap length: after an initial linear increase with length, the decay rate saturates. From the linear part, one can extract the effective trapping rate, which turns out to increase with increasing fridge temperature. This finding (as well as independent measurement of the relaxation rate) is in qualitative agreement with the expectation of slow relaxation in the normal metal being the bottleneck for trapping, see Eq. (113).

A more accurate modelling of the qubit geometry was considered in Ref. [32], where optimization in the number and position of traps was also studied, together with the other advantages of using traps (increase in the steady-state $T_1$ time and reduction of its fluctuations over

---

[2]in the regime $d \ll \lambda_{tr} \ll L$, for the slow modes one can neglect the second term in Eq. (116), which then reduces to the expression found in Ref. [31].

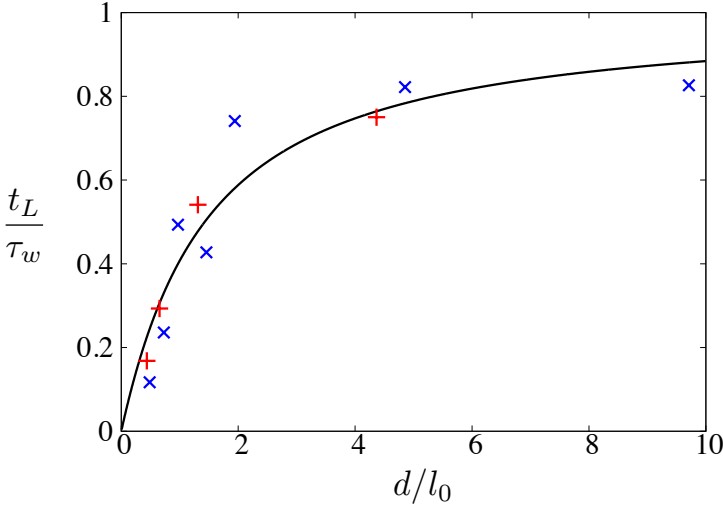

Figure 12: (from Ref. [31]) Quasiparticle density decay rate $s = 1/\tau_w$ (normalized by the inverse of the diffusion time $t_L = 4\tau_D/\pi^2$) vs trap size (normalized by $l_0 = \pi D_{\text{qp}}/2\Gamma_{\text{eff}}L$). The parameters $t_L$ and $l_0$ depend on temperature via the $T$-dependence of the quasiparticle diffusion constant $D_{\text{qp}}$ and $\Gamma_{\text{eff}}$ defined in Eq. (113). The blue"x" (red "+") symbols are for experiments performed at a fridge temperature of 13 mK (50 mK), at which the parameters $t_L \approx 184\,\mu s$ ($t_L \approx 125\,\mu s$) and $l_0 \approx 41\,\mu m$ ($l_0 \approx 46\,\mu m$) where estimated. The solid line is obtained from a numerical solution of Eq. (116) when neglecting the second term.

long time scales). It should be noted that normal-metal traps can also introduce new dissipation mechanisms for the qubit. For example, the inverse proximity effect broadens and soften the superconducting gap, introducing subgap states into which the qubit can loose energy; this effect weakens exponentially with the ratio between junction-trap distance over coherence length and can be neglected [33]. Current within the normal metal and the tunneling current through the barrier between the normal metal and superconductor can also dissipate energy. While the contribution of the latter to qubit relaxation is negligible, the former one imposes some constraints on trap design which are not relevant to current qubits, but could be limiting for qubits with improved coherence [34]. Such limitations of normal-metal traps can be largely sidestepped by using instead gap-engineered traps, obtained by good contacts between two different superconductors [35].

## 5.4 Quasiparticle trapping in Andreev levels

At the end of Section 2.2 we mentioned that the normal-state conductance scales proportionally to the product of the junction's cross-sectional area $\Sigma$ and the electron transmission coefficient $|t_B|^2$ of the tunnel barrier, $G_N \propto \Sigma \cdot |t_B|^2$. In discussing the Josephson effect in Section 2.3 and thereafter, we concentrated on the low-transmission, large-area tunnel junctions with $G_N \sim e^2/\hbar$. One may ask, if any new effects appear in smaller-area junctions, where the same value of $G_N$ is achieved by increasing $|t_B|^2$. The answer is affirmative, due to the increasing prominence of the sub-gap Andreev levels associated with a junction.

Phase biasing of a junction of any $|t_B|^2$ leads to the Andreev levels appearance. We may illustrate it with a simple example of a point contact with $|t_B|^2 \ll 1$. In terms of tunneling Hamiltonian (33), a contact of an area $\Sigma \ll \lambda_F^2$ is modeled by a matrix $t_{n_L n_R}$ having only one non-zero eigenvalue (i.e., only a single electron mode may go through the junction). Without

loss of generality, we may take $(\nu_0\mathcal{V})t_{n_L n_R} = t_B$ independent of $n_L, n_R$ [here $\mathcal{V}$ and $\nu_0$ are, respectively, the volumes of and the electron density of states in the two leads which we assume identical, cf. Eq.(40)]. At $|t_B| \ll 1$, we concentrate on a single-quasiparticle sector, keep only the term (44) and dispense with other terms (which do not conserve the quasiparticle number) in the tunneling Hamiltonian (43). Furthermore, expecting shallow bound states just below $\Delta$ at small $|t_B|$, we replace $u_{n_L} u_{n_R}^* - v_{n_L} v_{n_R}^*$ in Eq. (46) by $i\sin(\varphi/2)$ and therefore simplify Eq. (44) to:

$$\mathcal{H}_T^{\text{qp}} = i\frac{t_B}{\nu_0\mathcal{V}}\sin\left(\frac{\varphi}{2}\right)\sum_{n_L, n_R, \sigma}\gamma_{n_L\sigma}^\dagger \gamma_{n_R\sigma} + \text{h.c.}. \tag{118}$$

Now we perform a canonical rotation into a new quasiparticle basis defined by the operators $\gamma_{n\sigma\pm} = (1/\sqrt{2})(\gamma_{n_R\sigma} \pm i\gamma_{n_L\sigma})$. In new variables, the Hamiltonian for low energy ($\epsilon_n \approx \Delta + \xi_n^2/2\Delta$) quasiparticles in two identical leads linked by the junction takes the form

$$\mathcal{H} = \sum_{n,\sigma,\pm}\left(\Delta + \frac{\xi_n^2}{2\Delta}\right)\gamma_{n\sigma\pm}^\dagger \gamma_{n\sigma\pm} + \sum_{n,m,\sigma,\pm}\left[\pm\frac{t_B}{\nu_0\mathcal{V}}\sin\left(\frac{\varphi}{2}\right)\right]\gamma_{n\sigma\pm}^\dagger \gamma_{m\sigma\pm}. \tag{119}$$

In the continuum limit, we may replace the summation over $n$ here with integration over $\xi$, thus relaxing the requirement for the leads to be microscopically identical, $\sum_n\{\ldots\} \to (\mathcal{V}\nu_0)\int d\xi\{\ldots\}$. It is instructive to compare the resulting Hamiltonian of one of the fermion species (+ or −) with a Hamiltonian of free particles in one dimension subject to a potential $-U\delta(x)$ with $U > 0$, written in momentum representation:

$$\mathcal{H}_{\text{free}} = \int dp\,\frac{p^2}{2m}c_p^\dagger c_p + U\int dp\int dk\, c_p^\dagger c_k. \tag{120}$$

The comparison allows us to identify $1/m$ with $\mathcal{V}\nu_0/\Delta$ and $U$ with the "potential", either $+\sqrt{\mathcal{V}\nu_0}t_B\sin(\varphi/2)$ or $-\sqrt{\mathcal{V}\nu_0}t_B\sin(\varphi/2)$, for one of the species which has a negative potential at a given value of $\varphi$. A $\delta$-function well creates a localized state at any $U > 0$. Likewise, a localized Andreev state with energy

$$E_A(\varphi) = \Delta - 2E_J\sin^2(\varphi/2) = \Delta - E_J + E_J\cos\varphi \tag{121}$$

is formed at any $\varphi \neq 0$. In writing Eq. (121), we expressed the binding energy ($\propto |t_B|^2$) in terms of the Josephson energy (51) evaluated for the same parameters of tunneling Hamiltonian, $E_J = |t_B|^2\Delta/4$.

A remarkable property of the phase dispersion $E_A(\varphi)$ is that it is exactly opposite to the phase dispersion of the ground-state energy $\delta E_{GS}(\varphi)$, cf. Eq. (51). We derived it for a point contact with transmission coefficient $|t_B|^2 \ll 1$. In fact, this property is preserved for any value of the transmission coefficient, as long as (i) the time-reversal symmetry (at $\varphi = 0$) is preserved, and (ii) electrons acquire a negligible phase while traversing the junction. The latter condition is satisfied even for a ballistic point contact ($|t_B|^2 \to 1$) as long as its length is small compared to the superconducting coherence length. At arbitrary transmission,

$$E_A(\varphi) = \Delta\sqrt{1 - |t_B|^2\sin^2(\varphi/2)}. \tag{122}$$

This relation, and the respective modification of $\delta E_{GS}(\varphi)$, can be obtained in multiple ways, including a non-perturbative treatment of the tunneling Hamiltonian (43) and the use of scattering matrix formalism [36]. A short, wide-area junction can be viewed as a set of parallel quantum channels characterized by their respective transmission coefficients $|t_B^i|^2$. each of the channels creates an Andreev bound state with energy $E_A^i(\varphi)$ obtained from Eq. (122) by replacing $t_B \to t_B^i$. The phase-dependent part of the ground-state energy is modified compared to Eq. (51), $E_J(1 - \cos\varphi) \to -\sum_i E_A^i(\varphi)$.

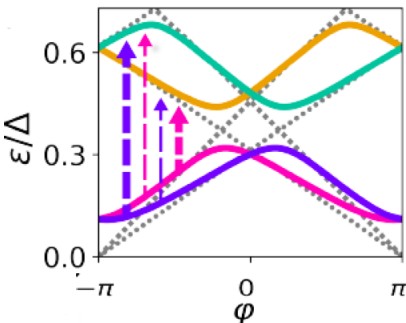
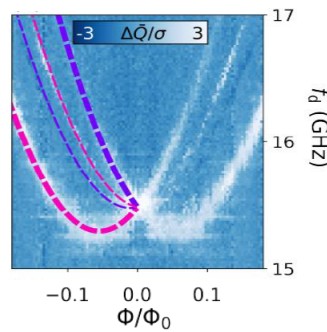

Figure 13: (from Ref. [37]) Left panel: a sketch of the Andreev levels dispersion with $\varphi$ in a single-mode highly transparent junction. Its length is assumed to somewhat exceed the superconducting coherence length, allowing for two Andreev levels. The level degeneracy at $\varphi = 0$ is the manifestation of time-reversal symmetry; the degeneracy at $\varphi = \pm\pi$ is due to the symmetry with respect to the product of time reversal and spatial inversion transformations. At other values of $\varphi$, Kramers doublets are split due to the combination of a finite Josephson current and present spin-orbit coupling. Various dashed arrows indicate transitions involving promotion of a quasiparticle from a lower to higher-energy Andreev state. Right panel: spectroscopic lines corresponding to the indicated transitions. The lines intersection at $\Phi/\Phi_0 = 0$ reflects the $\varphi = 0$ Kramers degeneracy.

The described spectrum of Andreev levels and their relation to the phase dependence of the ground state energy is specific for short junctions and requires time-reversal symmetry to be present at $\varphi = 0$. Violation of any of these two conditions modifies the Andreev levels and breaks down their relation to the ground-state properties. The phase of an electron wave function accumulated in the course of propagation through a finite-length junction reduces the energy of an Andreev level: its energy is $E_A < \Delta$ even at $\varphi = 0$. A longer junction may host more than one Andreev level; the levels retain their Kramers degeneracy, at least at $\varphi = 0$. Regardless the junction's length, Zeeman effect associated with an applied magnetic field breaks time-reversal symmetry and lifts the spin degeneracy even at $\varphi = 0$. A phase bias $\varphi \neq 0$ across the junction leads to a Josephson current and is another source of time-reversal symmetry breaking. In the presence of spin-orbit coupling, a finite Josephson current may cause the spin splitting of an Andreev level. Such spin-split structure of Andreev levels in a finite-length, high-transmission junction is sketched in the left panel of Fig. 13.

The Andreev levels lie below the edge of the quasiparticle continuum, and therefore are prone to trap quasiparticles. In terms of occupation factors $n_A^i$ of the Andreev levels, the corresponding correction to the energy of the junction is $\delta E(\varphi) = \sum_i n_A^i E_A^i(\varphi)$. At each given time, the set of factors $\{n_A^i\}$ is drawn from integers 0 and 1 (we assign different superscripts to the components of a degenerate level). The set $\{n_A^i\}$ changes from time to time, due to the inelastic relaxation of quasiparticles interacting with phonons. Based on Eqs. (101) and the discussion in the beginning of Section 5, we expect their rate to be slower than $1/\tau_N(\Delta)$. The rate is reduced further by low total average number of trapped quasiparticles, in which case we also may expect the energy relaxation to occur faster than the recombination.

At fixed $\{n_A^i\}$, the trapped quasiparticles affect the inductance of the junction. If the latter is a part of an $LC$-circuit, trapping shifts down its resonance frequency. This shift provides one with a measurable "fingerprint" of $\{n_A^i\}$. This kind of experiment was performed [38] with an Al nanobridge which may be considered as a short junction of a cross-section passing about

SciPost Phys. Lect.Notes 31 (2021)

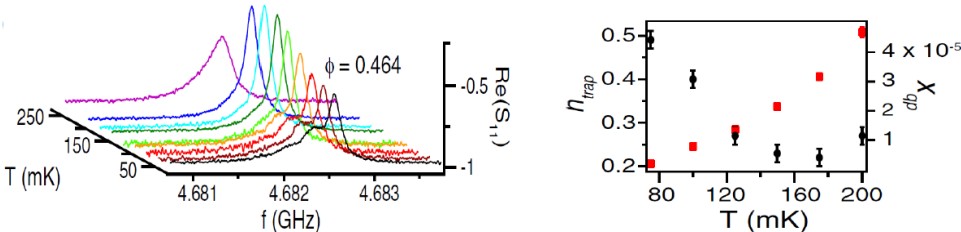

Figure 14: (from Ref. [38]) Left panel: resonance traces of an *LC* circuit containing an Al nanobridge. The estimate number of quantum modes propagating through the bridge is ∼ 700. Traces are averaged over many measurements spanning time interval exceeding the time needed for the rearrangement of the set of occupation factors $n_A^i$. At low temperatures, two shoulders are clearly seen on the low-frequency side of the main resonance. The shoulders correspond to the shifts of the resonance frequency caused by quasiparticle poisoning, respectively, of one or two Andreev states. Right panel: the average number of trapped quasiparticles (left axis, black circles) and $x_{qp}$ (right axis, red squares) extracted from experiment.

700 electron modes. Under applied phase bias, each mode with a particular value of $|t_B^i|^2$ gives rise to an Andreev level with energy $E_A^i$ given by Eq. (122). At a given bias $\varphi$, resonance traces of the reflection amplitude were accumulated over time exceeding the evolution time of the $\{n_A^i\}$ set. The averaged trace therefore corresponds to an average over all configurations $\{n_A^i\}$, weighted by their probabilities. As flux bias grows, energies $E_A^i(\varphi)$ drop, and so the likelihood of a quasiparticle trapping in an Andreev state grows. The left panel of Fig. 14 shows the averaged traces at $\varphi = 0.464\pi$ for a set of temperatures. At the lowest temperature, multiple shoulders on the low-frequency side of the main resonance are resolvable, indicating multiple quasiparticle trapping numbers. The most prominent shoulder corresponds to a single quasiparticle trapped by an Andreev level associated with one of the modes. Its width comes from the range of the $E_A^i(\varphi)$ values of ∼ 700 Andreev levels. It is quite remarkable that quasiparticle "poisoning" of one out of 700 quantum modes is traceable in the experiment. A weaker feature in the trace appearing further away from the main resonance peak corresponds to trapping of two quasiparticles. At higher temperatures, first the 2-quasiparticle and then the 1-quasiparticle shoulder shrink, leading to a Lorentzian resonance at $T \sim 150$ mK. The extracted from experiment temperature dependence of the average number of trapped quasi-particles $\overline{n}_{trap}$ is shown in Fig. 14. At $T \lesssim 170$ mK, the number $\overline{n}_{trap}$ grows as temperature is reduced; this is characteristic for a non-equilibrium population.

Microwave technique was recently also applied to studying Andreev levels in atomic point contacts [39, 40] and in proximitized semiconductor wires [37, 41–43]. Such junctions carry only one or a few electron modes which allows one to perform spectroscopy of individual levels. There is a simple rule of thumb for assessing the odds of quasiparticle poisoning of an Andreev level at low temperature $T$ and small, temperature-independent $x_{qp}$. Assuming Boltzmann distribution of the quasiparticles in energy, we find their chemical potential (measured from the edge of the quasiparticle continuum), $\mu = (T/2)\ln(x_{qp}^2 \Delta/2\pi T)$. The justification for the use of Boltzmann distribution is the inequality between the relaxation and recombination rates, $1/\tau_E \gg 1/\tau_r$. This, in turn, requires small occupation factors of the quasiparticle states. Once the energy $E_A(\varphi) - \Delta$ of an Andreev level drops below $\mu$, one may expect its high occupation. This condition was clearly satisfied in experiments [37], where all the detected transitions, see Fig. 13, had $n_A^i = 1$ in the initial state.

# 6 Conclusions

Constructing a quantum information device calls for finding elementary building blocks capable of maintaining quantum coherence over extended time periods. Superconductors provide one with a head-start in the race for a perfect device: the superconducting ground state locks together a macroscopic number of degrees of freedom, leaving a small number of collective variables to build a qubit from. The coherence distributed over many particles is inherently more robust than that of a single spin or atom. This robustness allows one to shorten the preparation and readout times for a superconducting qubit. However, some hazards come along with the macroscopic dimensions of a qubit. Many of them are defeated by now ubiquitous circuit QED architecture [44–46]. That sharpened the attention to the unwanted influences of superconducting quasiparticles in the "conventional" circuit QED devices [47] and in putative topological qubits [48]. It is clear by now that the observed low-temperature quasiparticle density by far exceeds its equilibrium values in a broad variety of devices. Their sources are still not fully identified, with photons [20], phonons [49,50], and even cosmic rays [47,51,52] being contenders. Meanwhile, superconducting qubits have provided one with an unrivaled technique for time-domain experiments. In many cases, it is by far the most sensitive tool for investigation of elementary processes in quasiparticle dynamics. It is this tool that allowed one to resolve such subtle effects as the tiny dissipative $\cos\varphi$-component of the Josephson current and quasiparticle trapping rate by a core of a single vortex line. Improving the qubit performance goes hand-in-hand with the ever-increasing capability of the techniques they provide for physics research.

# Acknowledgements

We thank M. H. Devoret, S. M. Girvin, R. J. Schoelkopf, and members of their research groups for numerous discussions and collaboration. We are grateful to M. Houzet, P. Kurilovich, V. Kurilovich, S. S. Pershoguba, and H. Hsu for their thoughtful reading of the manuscript and help with improving it. This work was supported by ARO grant W911NF-18-1-0212 and by the Alexander von Humboldt Foundation through a Feodor Lynen Research Fellowship (G.C.) and by DOE contract DE-FG02-08ER46482 (L.I.G.).

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
