# Peer review of "Bogoliubov Quasiparticles in Superconducting Qubits"

_SciPost Physics Lecture Notes, doi:SciPost Phys. Lect. Notes 31 (2021)_

## Round 2 · Referee Report · Anonymous (Referee 1) · 2021-5-30

Report

Based on lectures given within the Les Houches summer school series, the authors review the effects of quasiparticle excitations in superconducting qubits, including their interaction with the qubit and their contribution to qubit relaxation and dephasing.

I find the review clear and well-written, giving a good starting point for both students and scientists interested in understanding the role of quasiparticle dynamics in superconducting qubits and its effect on qubit decoherence and relaxation. I have a few comments and suggestions:

1. In the paragraph above Eq.~(29) the authors argue that at an average gate offset charge $\bar{n}_{g}=1/4, 3/4$, the transmon energy levels are insensitive to $e$-jumps. I believe that a quantitative discussion of the transmon energy levels and their dependence on $n_g$ may be beneficial for the average reader.

2. A few typos/grammatic:
*) Three lines below Eq.~(2) - "As the result,..." should be "As a result,...". This appears also in the last paragraph before Section 2, in the 8th line below Eq.~(30), and in the line below Eq.~(79).
*) Seven lines below Eq.~(2) - To be consistent with Eq.~(2), $(\lambda/\nu_0)\psi^{}_{k}(\mathbf{r})\psi^{}_{l}(\mathbf{r})\psi^{\ast}_{m}(\mathbf{r})\psi^{\ast}_{n}(\mathbf{r})$ should be $(\lambda/\nu_0)\psi^{\ast}_{k}(\mathbf{r})\psi^{\ast}_{l}(\mathbf{r})\psi^{}_{m}(\mathbf{r})\psi^{}_{n}(\mathbf{r})$.
*) Three lines below Eq.~(7) - $\omega^{}_{\mathrm{D}}/\delta\epsilon$ should be $\hbar\omega^{}_{\mathrm{D}}/\delta\epsilon$.
*) I think the minus sign on the right hand side of Eq.~(13) is redundant, because Eqs.~(6) and~(8) are defined without a minus sign. So either Eqs.~(6) and~(8) should be written with a minus sign (as conventionally done in BCS theory) or Eq.~(13) be written with a plus sign.
*) The equation for $T^{\ast}$ at the beginning of page 7 is not numbered.
*) Third line below Eq.~(46), "...comes from first term..." should be "...comes from the first term..."
*) Last line on page 19 - junction should be junctions.
\newline *) First line below Eq.~(94) - "This in..." should be "This is in...".

3. Below Eq.~(40) the phase $\varphi$ is defined as $2(\phi^{}_{L}-\phi^{}_{R})$, whereas below Eq.~(45) it is defined as $2(\phi^{}_{R}-\phi^{}_{L})$, which differs by a minus sign. In Section 2.4 it seems that the central phase $\varphi$ is again defined as $2(\phi^{}_{L}-\phi^{}_{R})$. I suggest to keep the definitions consistent or at least explaining the origin for differences.

4. If the unitary transformation discussed above Eq.~(34) is of the form $|\tilde{\psi}(t)\rangle=U(t)|\psi(t)\rangle$, then it seems to me that the correct form of Eq.~(34) should be $\mathcal{\tilde{H}}=U\mathcal{H}U^{\dagger}+i\hbar\frac{dU}{dt}U^{\dagger}$.

5. The paragraph above Eq.~(39): "One may worry that we applied a formalism developed for isolated islands..." is a bit unclear, and I suggest explaining this point in more detail, or referring the reader to suitable reference where this formal derivation can be found.

6. I suggest that authors define the quasiparticle current spectral density $S_{\mathrm{qp}}(\omega)$ right after Eq.~(65) and relate it to Eq.~(64). Then it is easier to understand how Eq.~(66) is derived.

7. Could the authors shortly derive the replacement~(83) in the presence of cavity photons? I believe the reader could benefit from further details about the coupling of the photon electric field to the superconducting phase.

Requested changes

Please see attached file

Attachment

---

## Round 3 · Author Response

We thank the Referee for the useful suggestions. We have implemented all of them.
-
Explicit expressions for the phase slip amplitude and the transmon anharmonicity are added to the text after Eq. (29); in addition, we refer the reader to Ref. 6 for a detailed description of the energy levels. We also added a symmetry argument to explain the insensitivity of the qubit spectrum to e-jumps at n_g=1/4 and 3/4.
-
All noticed typos have been fixed.
-
The expression after Eq.(45) was also a typo, which has been corrected.
-
We kept the form of the unitary transformation unchanged: it is correct and corresponds to redefining U as U^\dagger in the version given by the Referee.
-
We have clarified the technical point related to the derivation of Eqs. (39) and (40) and, in addition, provided a reference to an alternative derivation of Eq. (40).
-
We have introduced the definition of S_qp in Eq. (67).
-
After Eq. (86), we expanded the explanation of the way the photon electric field couples to the superconducting phase and provided the necessary references to the equations appearing earlier in the text.
We believe the current version is acceptable for publication.

---

## Round 3 · List of Changes



---

## Editorial Decision

published